# Temporal morphogen gradient-driven neural induction shapes single expanded neuroepithelium brain organoids with enhanced cortical identity

Anna Pagliaro [1], Roxy Finger [1], Iris Zoutendijk [1], Saskia Bunschuh[1], Hans Clevers [1,2,3,4], Delilah Hendriks [1,2,3,5] ✉ & Benedetta Artegiani [1,5] ✉

Pluripotent stem cell (PSC)-derived human brain organoids enable the study of human brain development in vitro. Typically, the fate of PSCs is guided into subsequent specification steps through static medium switches. In vivo, morphogen gradients are critical for proper brain development and determine cell specification, and associated defects result in neurodevelopmental disorders. Here, we show that initiating neural induction in a temporal stepwise gradient guides the generation of brain organoids composed of a single, self-organized apical-out neuroepithelium, termed ENOs (expanded neuroepithelium organoids). This is at odds with standard brain organoid protocols in which multiple and independent neuroepithelium units (rosettes) are formed. We find that a prolonged, decreasing gradient of TGF-β signaling is a determining factor in ENO formation and allows for an extended phase of neuroepithelium expansion. In-depth characterization reveals that ENOs display improved cellular morphology and tissue architectural features that resemble in vivo human brain development, including expanded germinal zones. Consequently, cortical specification is enhanced in ENOs. ENOs constitute a platform to study the early events of human cortical development and allow interrogation of the complex relationship between tissue architecture and cellular states in shaping the developing human brain.

Human brain development is unique in multiple aspects as compared to other mammals[1–3], but its study has long been difficult due to the lack of suitable model systems. The development of three-dimensional (3D) in vitro human brain models, so-called organoids or spheroids, derived from pluripotent stem cells (PSCs) has emerged as a powerful tool to study brain development and associated diseases[4–7]. PSCs can be induced into neural fate and to reflect various regions of the brain using a regimen of small molecules and growth factors throughout the differentiation process, thereby attempting to mimic the signaling pathway activities throughout development[8–18]. Brain organoids have been amply shown critical to reveal important aspects of human brain development[19–25]. Nonetheless, some biological features are only limitedly recapitulated in brain organoids, such as the massive expansion of neural progenitors typical of the developing human brain[5] and its

[1]The Princess Maxima Center for Pediatric Oncology, Utrecht, The Netherlands. [2]Hubrecht Institute, Royal Netherlands Academy of Arts and Sciences, Utrecht, The Netherlands. [3]Oncode Institute, Utrecht, The Netherlands. [4]Pharma, Research and Early Development (pRED) of F. Hoffmann-La Roche Ltd, Basel, Switzerland. [5]These authors jointly supervised this work: Delilah Hendriks, Benedetta Artegiani. ✉e-mail: d.hendriks@hubrecht.eu; b.a.artegiani@prinsesmaximacentrum.nl

overall organization[26]. In fact, within each brain organoid, a distinctive morphological feature is the uncontrolled and spontaneous development of multiple rosettes, representing individual neuroepithelium structures. This may cause within-and-between organoid heterogeneity and may affect reproducibility[26]. Additionally, this does not parallel in vivo brain organogenesis, where its development critically originates from a single neural tube. Indeed, recent efforts have aimed at mitigating the formation of multi-rosettes through the generation of single-rosette organoids, by employing either manual isolation of single rosette structures or using micro-patterning approaches[27–32]. Furthermore, the use of biomaterials or microdevices has also been exploited to guide the formation of enlarged or folded neuroepithelium structures in vitro[33,34]. Likewise, genetic approaches through *PTEN* knock-out can induce surface folding of brain organoids through enhanced expansion of the neural progenitors[35].

Mammalian brain development is a highly regulated process shaped by strict time- and space-dependent morphogen gradients[36–38], and associated dysregulation results in brain developmental defects. Recent efforts have indeed probed the use of genetic or synthetic morphogen gradient-based approaches to influence in vitro patterning outcomes, either for early germ layer patterning or for brain topography specification[39–41]. Creating a local source of BMP4 signaling using a microfluidic approach revealed important aspects of germ layer patterning[39]. In brain organoids, a sonic hedgehog gradient provided via an inducible genetic approach revealed its importance in patterning ventral forebrain fate[40]. Finally, a chip approach using morphogen-soaked beads directed the emergence of distinct dorso-ventral and anterior-posterior topography within brain organoids[41]. Here, we probed the effect of timed morphogen gradients during neural induction. We develop a straightforward approach based solely on a stepwise temporal medium gradient, which, strikingly, resulted in major morphological changes that ultimately generated well-specified cortical organoids displaying a self-organized single continuum of expanded neuroepithelium. We find that the presence of a prolonged gradient of TGF-β signaling is important in this process.

## Results

### A temporal neural induction gradient induces generation of expanded neuroepithelium organoids

The generation of brain organoids from PSCs relies on an initial neural induction, followed by expansion/differentiation and maturation phases[42]. These are typically induced by "all-at-once" switches to specific media that may vary in composition and timing depending on the specific protocol. Given that tight and time-controlled morphogen gradients underlie correct in vivo development, we reasoned that providing morphogen switches in a temporal and gradual (stepwise gradient) manner could influence brain organoid phenotypes.

After initial dissociation of feeder-free human embryonic stem cells (H1 hESCs), and reaggregation into embryoid bodies in stem cell medium, we employed dual SMAD inhibition[11,13,43,44] for cortical neural induction (NI), and either switched to NI medium in a sudden- or stepwise manner (Fig. 1a and Supplementary Fig. 1a). We rigorously used the same number of initial cells for both protocols. In the stepwise, gradual NI protocol, cells are exposed to a prolonged and decreasing gradient of the stem cell medium, while concomitantly providing a stepwise, gradual increase in NI medium. Thereafter, medium was switched to expansion medium containing EGF and FGF2, and later to maturation medium from day 25 onwards (containing amongst others Matrigel). The fate during formation of the brain organoids under these different protocols was monitored over time. While prior as well as during NI the forming organoids were indistinguishable, approximately 12–14 days after protocol initiation, we observed a divergent morphological phenotype between organoids formed in the different conditions (Fig. 1b, c). Upon sudden NI, cortical

organoids (COs) formed with a typical spherical shape with multiple rosettes visible under brightfield microscopy. Instead, under the stepwise gradual NI, organoids adopted a convoluted shape that became more pronounced with time (Fig. 1b and Supplementary Fig. 2a). At day 14, a clearly distinct lighter border with ridges and folds at the apex of the organoids was visible, which became more prolonged and pronounced at day 24 (Fig. 1b and Supplementary Fig. 2a), suggestive of expanded neuroepithelium structures.

We measured morphological features of the organoids under the two conditions during a 25-day time-course, which confirmed a strikingly reduced circularity under gradual NI, already evident at day 14, and progressively decreasing to 0.5 at day 25 (Fig. 1d). Furthermore, upon appearance of these folded structures, the gradual NI-generated organoids displayed an increased organoid perimeter (Fig. 1e). Of note, organoid areas were slightly, yet significantly, increased at day 20 and day 25 under stepwise NI (Supplementary Fig. 1g). Altogether, these morphological changes and parameters were consistent across multiple organoid batches formed under stepwise NI versus sudden NI (Supplementary Fig. 2a–c).

As an additional comparison, we generated organoids using a commercially available protocol for dorsal forebrain organoids (CommOs) (Supplementary Fig. 1b). As expected, also in this case the organoids formed with a typical spherical morphology (Supplementary Fig. 1c), and with morphological parameters analogous to the COs formed under the sudden NI protocol (Supplementary Fig. 1d, e). Clearly, the organoids generated under the gradual NI were morphologically very distinct (Fig. 1b and Supplementary Figs. 1c, 2a). We next analyzed the cellular organization of the different organoids by immunostaining using the neuroepithelium marker N-Cadherin (NCAD). At 16- and 24-days, COs as well as organoids generated using the commercial protocol, consisted of NCAD+ cells organized as a collection of multiple neural rosettes, displaying various shapes, some more elongated and some more circular and of different sizes (Fig. 1f, g and Supplementary Fig. 1f). Instead, the temporal gradient-generated organoids formed an elongated, continuous, radially organized, and often folded NCAD+ neuroepithelium, resembling a ventricular zone (VZ)-like structure (Fig. 1f, g and Supplementary Fig. 1f, Supplementary Videos 1, 2). NCAD+ cells of the neuroepithelium observed in these organoids were located on the outside of the organoid, suggesting an apical-out morphology (Fig. 1f, g and Supplementary Fig. 1f). We sometimes observed organoids in which, in addition to the extended neuroepithelium, a few rosette-like or spherical structures would additionally form (Supplementary Fig. 3a–c).

Despite the major cellular organization changes and the overall distinct organoid architecture, qPCR analysis confirmed the absence of off-lineage markers analogous to the conventional CO (sudden NI) protocol, while their neural identity was confirmed by robust NCAD and Nestin expression (Supplementary Figs. 1h, 2d). Given the appearance of these expanded neuroepithelium structures in contrast to the conventional rosette-like neuroepithelia, we named these organoids generated with the temporal stepwise NI gradient ENOs (Expanded Neuroepithelium Organoids). To test the robustness of ENO formation, we employed two additional widely used hESC lines (H9 and H14). Again, while sudden NI generated the typical, spherical rosette-containing COs (Supplementary Fig. 4a, b), the temporal stepwise NI gradient resulted into successful generation of expanded neuroepithelium structures (Fig. 1c, d and Supplementary Fig. 4a, b), across different batches using the H9 and H14 lines, similar to H1 ENOs (Fig. 1h). This was further corroborated by the notable differences in organoid circularity and the organoid perimeters in H9 and H14 ENOs as compared to the respective CO controls of each line, resembling our previous observations using H1 hESCs (Fig. 1d, e and Supplementary Fig. 4c–f). Finally, NCAD staining of H9 and H14 ENOs revealed similar cellular organization and tissue architecture as observed for H1 ENOs (Supplementary Figs. 1i, 3a–c).

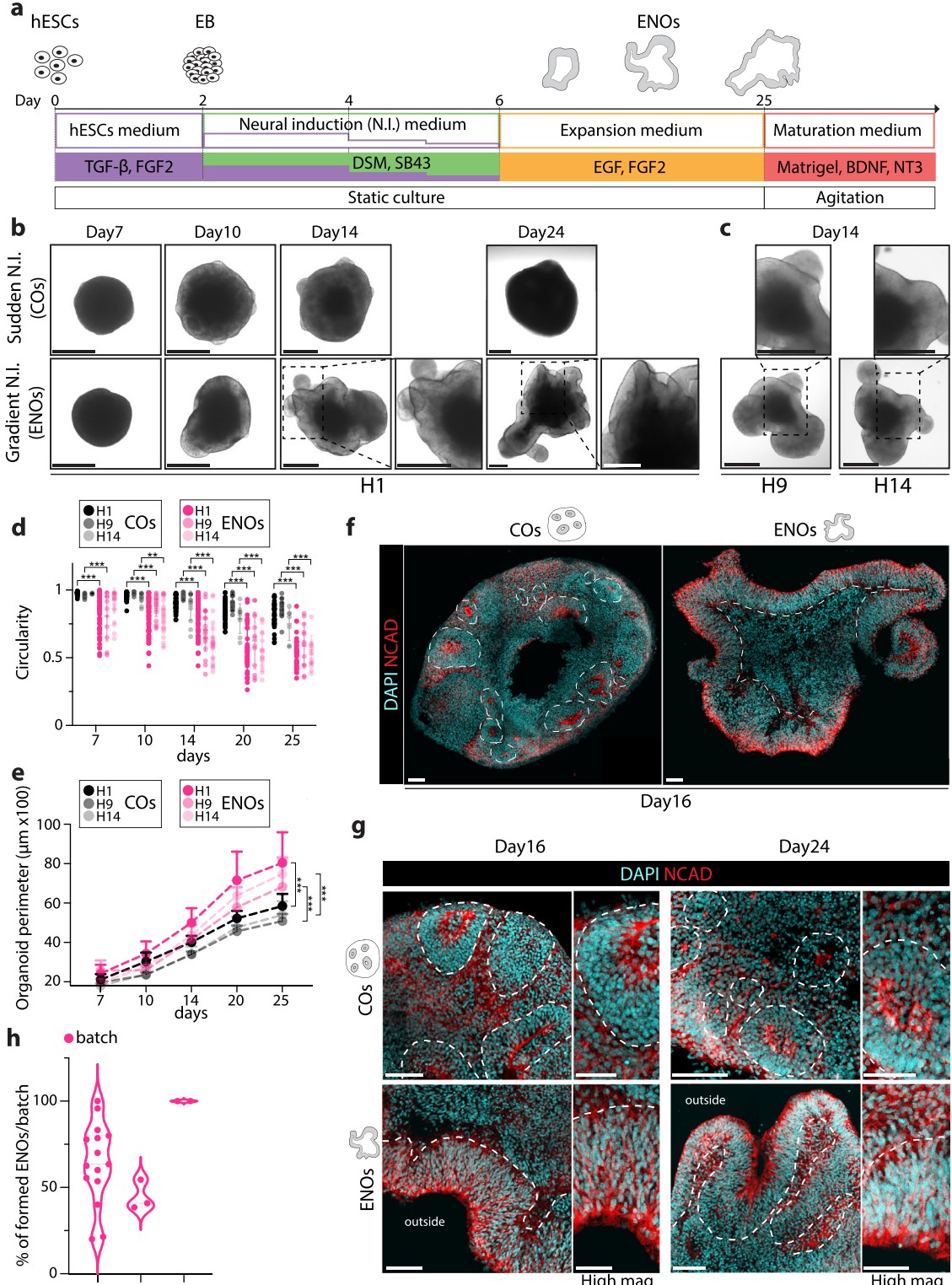

## TGF-β signaling gradient plays a major role in determining ENO formation

To understand which signaling pathway could be involved in the formation of ENOs, we carefully considered the composition of both the stem cell medium, which is gradually reduced during NI in a stepwise fashion, and of the NI medium which is instead gradually increased (Fig. 1a). FGF2 and TGF-β are the main morphogens present in the stem cell medium and are both needed for the maintenance of an undifferentiated embryonic stem cell state[45,46]. Our NI medium is instead based on a classical dual SMAD inhibition approach[43], in which the use of the TGF-β inhibitor (SB-431542, SB43) and the BMP inhibitor (dorsomorphin, DSM)[43,47,48] are provided during early stages of organoid formation to specify dorsal fate[11–13,44]. Given the concomitant decreasing gradient of TGF-β as well as the increasing TGF-β inhibitor concentration during neural induction in ENOs, we evaluated whether such controlled TGF-β gradient was responsible for the observed drastic

**Fig. 1 | Generation of Expanded Neuroepithelium Organoids (ENOs) through neural induction gradient modulation. a** Schematic illustration of timeline and protocol for generating ENOs (hESCs = human embryonic stem cells, EB = embryoid body, ENOs = expanded neuroepithelium organoids). **b** Representative brightfield images of H1 cortical organoids (COs−sudden N.I.) and H1 expanded neuroepithelium organoids (ENOs−gradient N.I.) at the indicated time points. **c** Representative brightfield images of H9 and H14 expanded neuroepithelium organoids (ENOs−gradient N.I.). **d** Quantification of the organoid circularity of COs and ENOs formed with H1, H9, and H14 hESC lines, measured at the indicated time points. Each dot represents an organoid and mean ± SD is plotted. \*\*\**p* < 0.001; Two-tailed unpaired *t*-test. **e** Quantification of the whole organoid perimeter of COs and ENOs formed with H1, H9, and H14 hESC lines, measured at the indicated time points. Mean ± SD is plotted. \**p* < 0.05; \*\*\**p* < 0.001; Two-tailed unpaired *t*-test.

**f** Representative fluorescence images of a whole CO and ENO stained for NCAD. Dashed lines delineate the basal side of the rosettes and neuroepithelium structures in the different organoids. **g** Representative immunofluorescence images of COs and ENOs stained for NCAD. Dashed lines delineate the basal side of the rosettes and neuroepithelium structures in the different organoids. **h** Quantification of ENO formation efficiency across multiple batches using H1, H9, or H14 hESC lines. Each dot represents a different batch. For **d**, **e**, a detailed description of how many organoids and batches were analyzed is described in Supplementary Table 3. Scale bars for **b**, **c** = 500 μm; **f** = 200 μm; **g** = 100 μm (low mag) and 50 μm (high mag). Images in **f** and **g** are representative of *n* = 3 independent experiments. Exact sample size and exact *P* values are provided in Source Data. Source data are provided as Source data file.

phenotype of ENOs. We, therefore, generated organoids under different levels of TGF-β signaling gradient modulation during NI from starkest-to-lowest inhibition over time (Fig. 2a, b): (1) the conventional COs, in which TGF-β is suddenly removed and replaced by its inhibitor SB43, therefore driving the most abrupt switch-off of TGF-β signaling; (2) a "full SB43" protocol, in which TGF-β in the stem cell medium is still decreased in a step-wise fashion, but the full inhibition by SB43 is added "all-at-once" on the first day of the neural induction, therefore still providing a more abrupt inhibition of TGF-β signaling; (3) the ENOs in which there is a period of concomitant decreasing gradient of TGF-β as well as an increasing gradient of SB43 concentration; and (4) a "no SB43" protocol, in which the concentration of TGF-β is decreased step-wise but is not counteracted by SB43, implying that its presence persists the longest in culture and therefore has the longest active signaling gradient. Considering day 0 to have the maximum presence of TGF-β (100%), all tested protocols start with the same concentration (Fig. 2a).

Corroborating our hypothesis that gradual TGF-β signaling modulation during NI is important in ENO formation, the no SB43 protocol also resulted in the formation of organoids that were phenotypically identical to the ENOs (Fig. 2b), and with high efficiency, an observation that was consistent across all 3 hESC lines tested, using for each multiple batches (Fig. 2c and Supplementary Fig. 4a–f). Conversely, the more abrupt inhibition of TGF-β signaling by the full addition of SB43 led to organoids which were morphologically much more similar to COs (Fig. 2b). When measuring the organoids' perimeter and circularity, no SB43 organoids were similar to ENOs, which was consistent across the H1, H9, and H14 lines (Fig. 2d, e and Supplementary Fig. 4c–f), while organoids generated with the full SB43 protocol resulted in organoids resembling COs (i.e., significantly different from ENOs) (Fig. 2d, e). The same similarities and differences were observed when the various organoids were evaluated by immunostaining. While the full SB43 organoids formed typical rosette structures, as visualized by NCAD staining on whole organoids (Fig. 2f and Supplementary Fig. 5a, b), the no SB43 organoids on the contrary formed the elongated apical-out neuroepithelium similar to what is observed in the ENOs (Fig. 2f and Supplementary Fig. 5a, b). These data position the TGF-β gradient as an important factor in ENOs establishment.

## ENOs display an extended phase of stem cell expansion
Given the major morphological and cellular organization changes, we performed more in-depth characterization of cell identity and developmental dynamics in the ENOs. The ENO structures were radially organized, and, similar to the cells in the rosettes in the COs, positive for the neural progenitor markers NCAD+ and SOX2+ (Figs. 1f, g, 3a, b, and Supplementary Fig. 5a, b). Abundant proliferating Ki-67+ cells were present in the VZ-like areas (Supplementary Fig. 6a). Cells expressing the intermediate progenitor marker TBR2 were found interspersed in the VZ (SOX2+) and created a denser layer above it, resembling an SVZ-like area (Fig. 3c). Additionally, newly generated neurons (born in the germinal layers and then migrating on top of them) were indeed visible

above the SOX2+ germinal areas, both in the ENOs and in the COs, as visualized by staining for the newborn neuronal marker DCX or the pan-neuronal marker TUJ1 (Fig. 3a, b and Supplementary Fig. 5c).

Prompted by the observed morphological features suggestive of an expanded neuroepithelium in the ENOs, we next assessed if developmental dynamics (i.e., timing of differentiation and neurogenesis) would be affected during organoid development. To this end, we used SOX2 as pan-progenitor marker to assess changes in stem cell abundance. Of note, the dorsal forebrain identity of these SOX2+ cells was confirmed by overlap for FOXG1 positivity in both ENOs and COs (Fig. 4a and Supplementary Fig. 8a). Interestingly, the percentage of SOX2+ cells over DAPI was significantly higher in the ENOs both at day 16 and day 24 as compared to COs (Fig. 3d). Moreover, the proportion of the organoid occupied by VZ-like regions, defined as radially organized NCAD+ areas composed of SOX2+ stem cells, was more pronounced in the ENOs and decreased more slowly overtime (Fig. 3g), suggestive of an extended stem cell expansion phase. As expected, SOX2+ cells gradually diminished in both ENOs and in COs during later timepoints (Fig. 3a, b, d), until a few SOX2+ cells were present in both ENOs and COs in areas reminiscent of the germinal zones visible in day 85-old organoids (Fig. 3a, b, d, f, Supplementary Fig. 6d).

The amount and localization of SOX2+ cells in the ENOs at later time points suggested that upon maturation the expanded neuroepithelium transition from a singular continued structure to smaller units of stem cells organized in a rosette-like fashion (Fig. 3f, Supplementary Fig. 6b–d), presumably caused by the addition of Matrigel. Indeed, until day 25, ENOs display an inverted polarity as compared to the COs, where the outside of the ENOs is NCAD/ZO-1 positive (Fig. 1f, g, Supplementary Fig. 7a). We also investigated the effects of Matrigel addition at different timepoints and concentrations during ENO formation. Addition at early timepoints (day 7 and 10) prevented ENO formation (Supplementary Fig. 7b–e), generating CO-like rosette-containing organoids, while addition of low concentrations of Matrigel at day 18, during the ENO expansion phase, induced a marked polarity switch, including the formation of large concave lumens within the organoid, as well as areas of larger expanded neuroepithelium structures (Supplementary Fig. 7f–i), corroborating that tweaking the timing and amount of Matrigel addition influences ENO structures and polarity (Supplementary Fig. 8). This is in line with recent findings in which exogenous extracellular matrices were shown to influence human telencephalic organoid formation[49].

The temporal SOX2 analyses suggested that ENOs display a delayed transition to neurogenesis. To probe this further, we quantified neurogenesis by assessing the amount of DCX+ newborn neurons over time (Fig. 3a, b). This analysis indeed revealed a reduced amount of DCX+ cells in ENOs compared to COs at early time points (days 16 and 24) (Fig. 3e). At the latest time points analyzed (days 50 and 85), the ENOs reached similar amount of DCX+ neurons as compared to the COs. Similarly, the amount of SOX2+ cells was equal in COs and ENOs at later time points (Fig. 3d). Accordingly, older ENOs displayed abundant presence of various types of cortical neurons (SATB2, CTIP2,

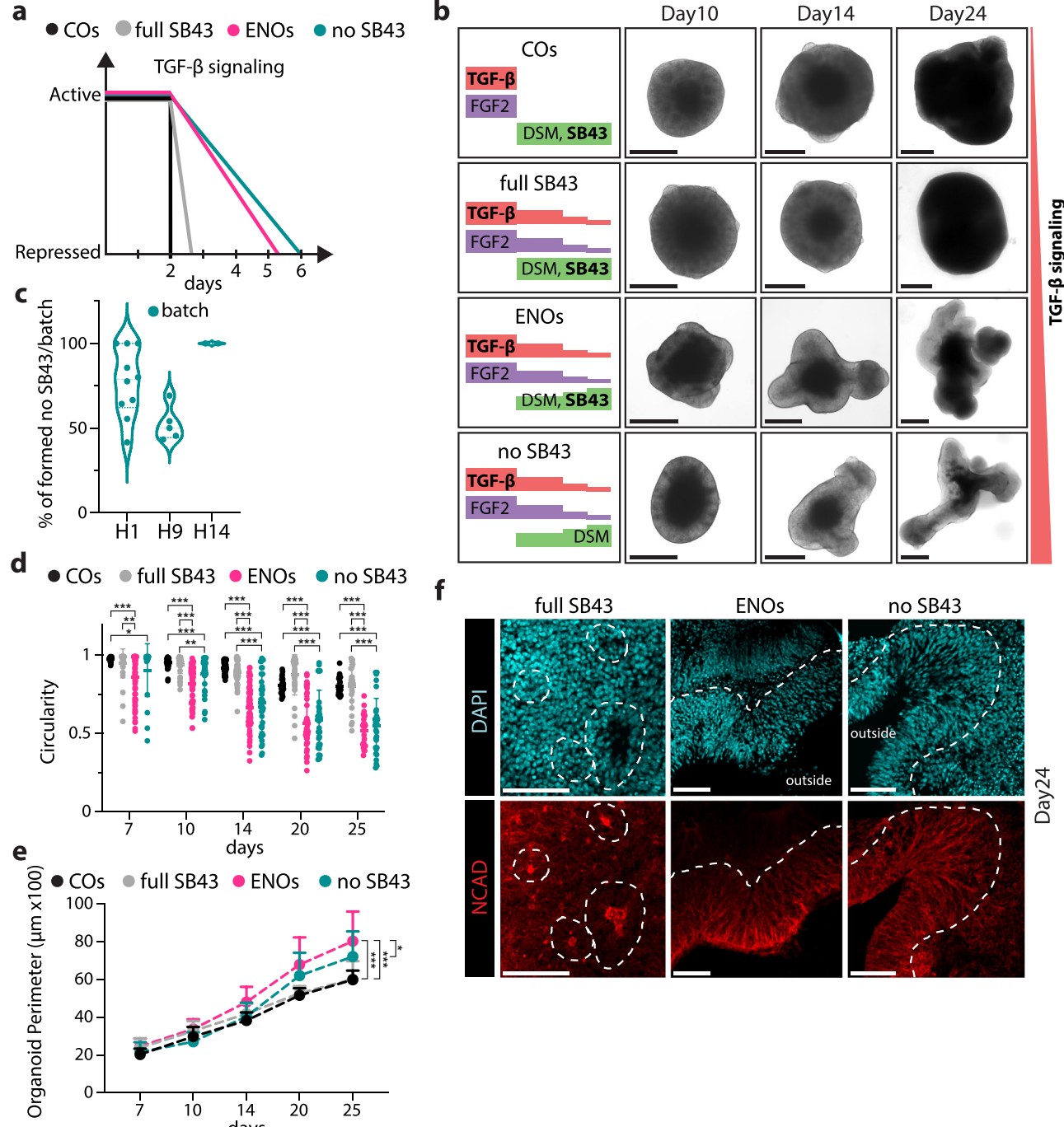

**Fig. 2 | TGF-β signaling gradient plays a major role in determining ENO formation. a** Schematic illustrating hypothetical TGF-β signaling strength throughout the first 6 days in culture of the compared conditions: COs, full SB43, ENOs, and no SB43. **b** Representative brightfield images of COs, full SB43, ENOs, and no SB43 conditions formed with the H1 hESC line at the indicated time points with on the left a graphical description of removal and addition of relative morphogens and small molecule inhibitors. Scale bars = 500 μm. **c** Quantification of ENO formation efficiency under no SB43 conditions across multiple batches using the H1, H9, or H14 hESC lines. Each dot represents a different batch. **d** Quantification of the organoid circularity of COs, full SB43, ENOs, and no SB43 measured at the indicated time points. Each dot represents an organoid and mean ± SD is plotted.

*$p < 0.05$; **$p < 0.01$; ***$p < 0.001$; Two-tailed unpaired $t$-test. **e** Quantification of the organoid perimeter of COs, full SB43, ENOs, and no SB43 formed with the H1 hESC line measured at the indicated time points. Mean ± SD is plotted. *$p < 0.05$; ***$p < 0.001$; Two-tailed unpaired $t$-test. **f** Representative immunofluorescence images of full SB43, ENOs, and no SB43 stained for NCAD. Dashed lines delineate the basal side of the rosettes and neuroepithelium structures in the different organoids. Scale bars = 100 μm. For **d**, **e**, a detailed description of how many organoids and batches were analyzed is described in Supplementary Table 3. Images in **b** and **f** are representative of $n = 3$ independent experiments. Exact sample size and exact $P$ values for **c**–**e** are provided in Source Data. Source data are provided as Source data file.

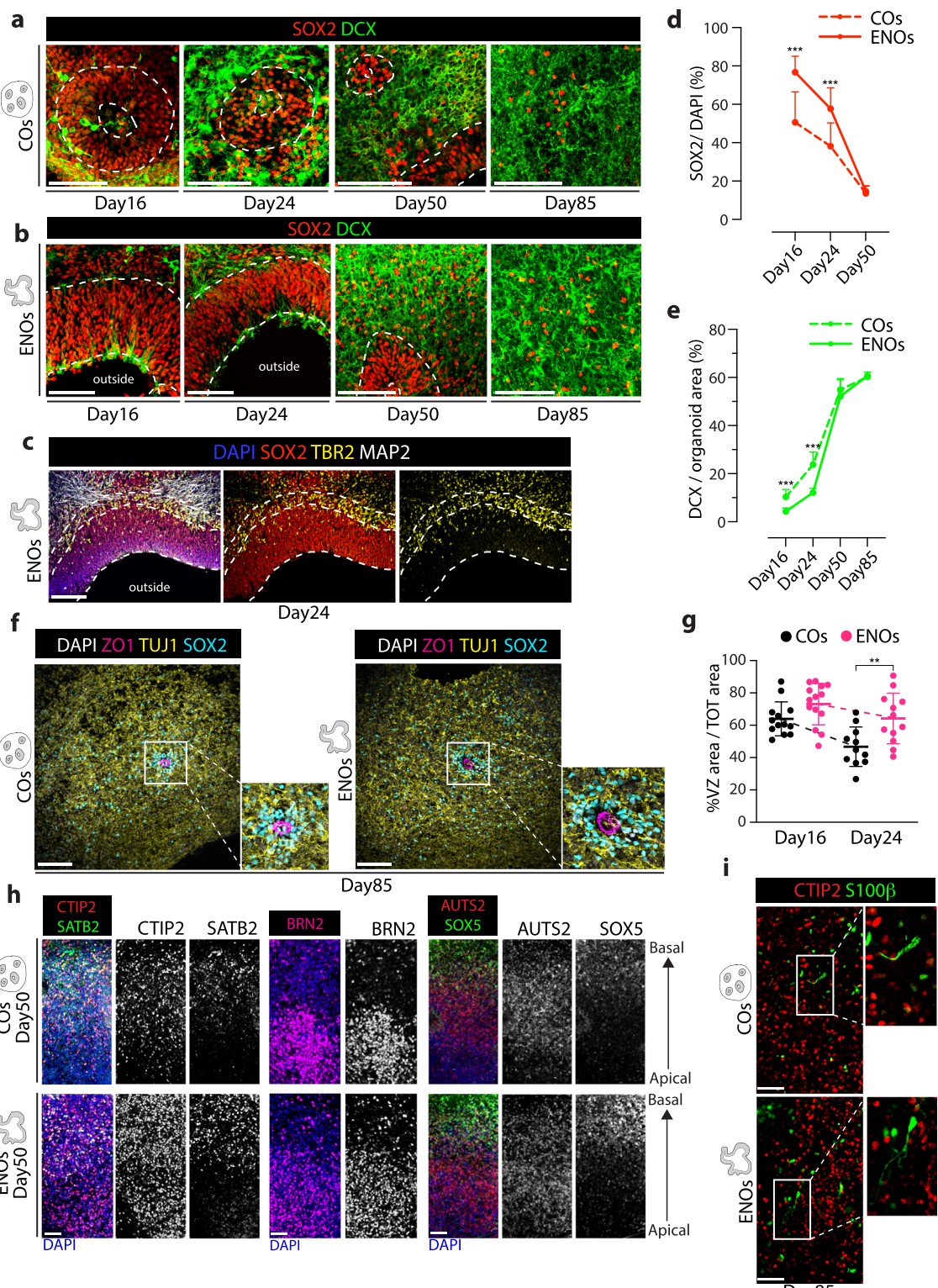

**Fig. 3 | ENOs display an extended phase of stem cell expansion.** Representative immunofluorescence images of COs (**a**) and ENOs (**b**) stained for SOX2 and DCX at the indicated time points. **c** Representative immunofluorescence image of ENO stained for TBR2, SOX2, and MAP2. Dashed lines delineate the VZ and SVZ-like regions of the neuroepithelium structure. **d** Quantification of the percentage of SOX2 positive cells over DAPI in ENOs (solid lines) and COs (dashed lines) in NCAD+ areas. Mean ± SD is plotted. ***$p < 0.001$; Two-tailed unpaired $t$-test. **e** Quantification of DCX as percentage over area in ENOs (solid lines) and COs (dashed lines). Mean ± SD is plotted. ***$p < 0.001$; Two-tailed unpaired $t$-test. **f** Representative immunofluorescence images of COs and ENOs stained for ZO-1,

SOX2 and TUJ1. Zoom-ins highlight rosette-like structures present in both COs and ENOs at this timepoint. **g** Quantification of the area occupied by the VZ over the whole organoid area in COs and ENOs. Measurements were performed over the whole area of the organoid based on NCAD+ neuroepithelium. ***$p < 0.001$; Two-tailed unpaired $t$-test. **h** Representative immunofluorescence images of COs and ENOs stained for CTIP2, SATB2, BRN2, AUTS2, and SOX5. **i** Representative immunofluorescence images of COs and ENOs stained for CTIP2 and S100β. Scale bars for **a**–**c** and **f** = 100 μm; **g**, **h** = 50 μm. Images in **a**, **b**, **f**, and **h** are representative of $n = 3$ independent experiments. Exact sample size and exact $P$ values for **d**, **e**, and **g** are provided in Source Data. Source data are provided as Source data file.

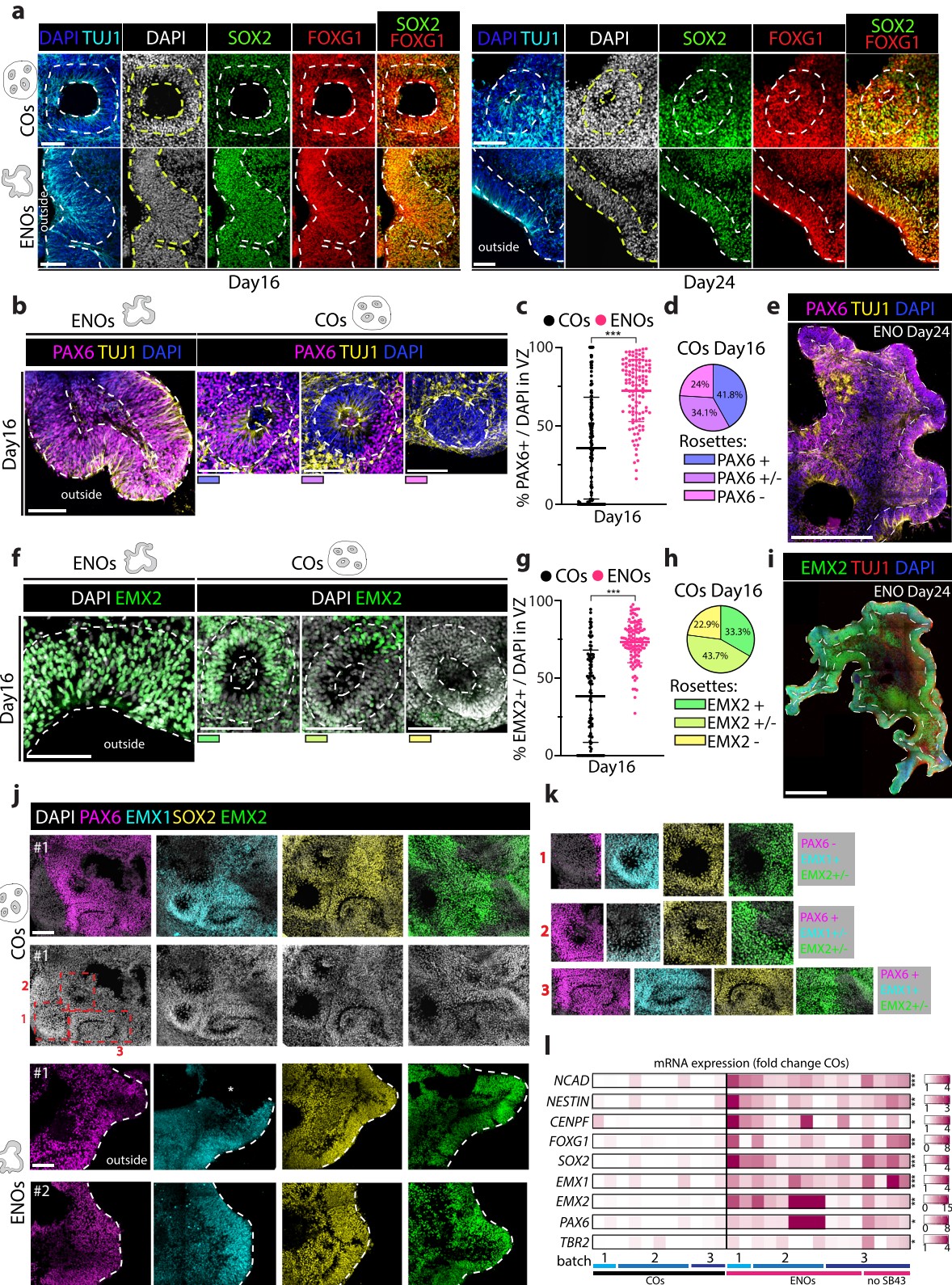

BRN2, AUTS2, SOX5) in an organized fashion (Fig. 3h), similar to what has been previously reported in multiple protocols of cortical organoids[11,18,50]. The abundance and layering were also akin to the neuronal organization observed in our COs (Fig. 3h). A few cells positive for the astrocytic marker S100β were present, interspersed within the CTIP2+ neurons, and in comparable amount both in ENOs and COs after day 50 (Fig. 3i).

Altogether, these data suggest that the ENOs undergo neuronal differentiation after an extended stem cell expansion phase and that this extended expansion phase does not affect the ultimate fate of the stem cells to produce neurons and glial cells with similar lamination patterns as in COs. Importantly, a prolonged and increased presence of progenitors is a distinguishing feature observed in human brain development as compared to other mammals[51–54].

**Fig. 4 | ENOs display enhanced cortical specification. a** Representative immunofluorescence images of COs and ENOs stained for TUJ1, SOX2, and FOXG1. **b** Representative immunofluorescence images of ENO and COs stained for PAX6 and TUJ1. **c** Quantification of the percentage of PAX6+ cells over DAPI in VZ of COs and ENOs. ***p < 0.001; Nonparametric Mann–Whitney U test. **d** Pie chart representing the percentages of homogeneously PAX6 positive (PAX6+), negative (PAX6−) and mixed (PAX6+/−) rosettes in COs. Sample size as in **c. e** Representative immunofluorescence image of a whole ENO stained for PAX6 and TUJ1. **f** Representative immunofluorescence images of ENOs and COs stained for EMX2. **g** Quantification of the percentage of EMX2+ cells over DAPI of COs and ENOs. ***p < 0.001; Nonparametric Mann–Whitney U test. **h** Pie chart representing the percentages of homogeneously EMX2 positive (EMX2+), negative (EMX2−) and mixed (EMX2+/-) rosettes in COs. Sample size as in **g. i** Representative immunofluorescence image of a whole ENO stained for EMX2 and TUJ1. **j** Representative

immunofluorescence staining of consecutive sections of day 16 COs and ENOs stained for PAX6, EMX1, SOX2, and EMX2. *=broken section. **k** Magnification of the numbered rosettes in COs (see **j**), displaying differences in PAX6, EMX1 and EMX2 positivity and co-expression patterns. **l** mRNA expression analysis of selected markers in ENOs and no SB43 organoids relative to COs at day 16 (normalized per batch). Each box represents an organoid. *p < 0.05, **p < 0.01, ***p < 0.001 comparing ENOs and no SB43 versus COs, ns = not significant; Two-tailed unpaired t-test. Scale bars for **a, b, f,** and **j** = 100 μm; **e** and **i** = 500 μm. Dashed lines in **a, b, e, f,** and **i** delineate apical and basal side of rosettes and neuroepithelium structures. For **c** and **g** each dot represents quantification in a different area of each measured rosette/neuroepithelium structure within the VZ and mean ± SD is indicated. Images in **a, b, f, i,** and **j** are representative of n = 3 independent experiments. Exact sample size and exact P values for **c, g,** and **l** are provided in Source Data. Source data are provided as Source data file.

## ENOs display enhanced cortical specification

We next questioned whether the major changes observed in the ENOs would also translate to differences in cell identity specification. First, we verified the telencephalic identity of the ENOs and the COs by co-staining SOX2 with FOXG1. This revealed that in both conditions SOX2+ areas are largely also FOXG1+ (Fig. 4a and Supplementary Fig. 9a), confirming the telencephalic identity. When compared to the cortical organoids (both COs and CommOs), ENOs displayed instead several differences concerning the expression of dorsal telencephalic fate markers in the SOX2+/FOXG1+ stem cell areas. Concomitant expression of PAX6 and EMX2 in radially organized areas in brain organoids is a measure of cortical identity specification[18,38]. Both the rosettes in the cortical organoids as well as the neuroepithelium structures in the ENOs were found positive for the cortical progenitor markers PAX6 and EMX2, but to a very different extent (Fig. 4b–k and Supplementary Fig. 10a–n). The ENOs displayed a significantly higher percentage of PAX6+ cells over total cells in the VZ at the different time points analyzed (day 16 and day 24) (Fig. 4b, c and Supplementary Fig. 10a–i). Additionally, while the expression of PAX6 was rather homogeneous and present in the germinal areas of the expanded neuroepithelium in the ENOs (Fig. 4b, e and Supplementary Fig. 10a, Supplementary Videos 1–2), PAX6+ cells were more scattered in the rosette-shaped neuroepithelia developed in cortical organoids, sometimes confined to one side of the rosette, and we also observed fully PAX6- rosettes (Fig. 4b–d and Supplementary Fig. 10a–i). These differences in PAX6 expression between ENOs and COs were robust across different batches (Supplementary Fig. 10d, e). Moreover, a similar homogenous expression of PAX6 was observed in the neuroepithelium structures of H9 and H14 ENOs, while conversely, rosettes in H9 and H14 COs displayed the same varied expression of PAX6 as already observed in H1 COs (Supplementary Fig. 11a–c). Comparable expression patterns and abundancy differences in COs and ENOs were also observed for the cortical marker EMX2 (Fig. 4f–k and Supplementary Fig. 10j–l), again reproducible across different batches (Supplementary Fig. 10m, n).

These observations were suggestive of improved cortical specification of the ENOs, which we further validated by staining of serial sections to assess the presence of co-positive regions for EMX2, PAX6 as well as EMX1, an additional cortical marker. In COs, SOX2+ rosettes often did not display overlapping co-expression of EMX2, PAX6 and EMX1 (Fig. 4j, k and Supplementary Fig. 9b). Similar to EMX2 and PAX6, also not all rosettes in COs stained positive for EMX1 (Fig. 4j, k and Supplementary Fig. 9c). In contrast, in the ENOs, the neuroepithelium areas displayed consistent and co-occurring expression of all these markers, both at day 16 and day 24 (Fig. 4j and Supplementary Fig. 9d). Finally, to corroborate our findings, we also assessed a panel of markers on mRNA level, comparing the expression between COs and ENOs across different batches. We included general stem cell markers (*NCAD, NESTIN, SOX2*), the general telencephalic marker *FOXG1*, the cell cycle-related marker *CENPF*, as well as various dorsal forebrain progenitor

markers (*TBR2, EMX1, EMX2, PAX6*). This analysis showed consistent expression (Supplementary Fig. 2d) and upregulation of these markers in the ENOs as compared to the COs (Fig. 4l), corroborating their enhanced stem cell expansion as well as their increased cortical identity.

## ENOs display tissue architecture features more closely recapitulating human brain development

Changes in epithelial features and cellular morphology of neuroepithelial cells have been described as one of the causes for the differences in growth observed very early on during human- and ape-derived brain organoid formation[54]. Prompted by their prominent architectural alterations at whole organoid level, we further investigated the morphological features of the expanded neuroepithelium structures in the ENOs at cellular level. Using ZO-1 to identify the apical side of the neuroepithelium in whole ENOs and COs (Fig. 5a, b and Supplementary Fig. 12a, b, Supplementary Video 2), we measured the apical perimeter of NCAD+ structures in both ENOs and COs at day 16 and 24. This analysis revealed a clear difference in shape and size of the apical surface between the two conditions and confirmed the apical-out nature of the ENOs. While within all COs (and CommOs) there were multiple lumina of relatively round shape, in each ENO there was a long and expanded apical continuum (Fig. 5a–c and Supplementary Fig. 12a–c). Additionally, while the apical perimeter of rosettes clearly decreased between day 16 and day 24 in COs (ca. 50%), this decrease was not observed in neuroepithelium structures of the ENOs (Fig. 5c and Supplementary Fig. 12c). Measurement of the surface area of cells on the apical side of the VZ-like structures revealed that ENOs display stem cells with an apical surface area significantly bigger as compared to the cells in the rosettes formed in COs (Figs. 5b, d, Supplementary Fig. 12b) at both time points. Interestingly, increased cellular apical surface area has been previously linked to a delayed transition to neurogenic radial glia, a characteristic of human brain development[54]. We further investigated the thickness of the VZ, by measuring the length from the apical side of the neuroepithelium, positive for NCAD and ZO-1, until the appearance of the first TUJ1+ neurons (Fig. 5e and Supplementary Fig. 12d). These defined germinal zones consisted of SOX2+ neural stem cells (Fig. 5e and Supplementary Fig. 12e). The neuroepithelium of the ENOs was significantly thicker than in the rosette structures of cortical organoids, possibly due to the delayed onset of neurogenesis found in the ENOs, and it reduced overtime from day 16 till day 24, likely due to ongoing neurogenesis (Fig. 5f and Supplementary Fig. 12f).

Similar differences in apical perimeter and VZ thickness were observed when comparing the ENOs to the condition with a more abrupt inhibition of TGF-β signaling (full SB43) (Fig. 2a, b). In this condition, the apical perimeter and VZ thickness were comparable to the ones observed in the rosettes of the COs (Supplementary Fig. 5c–g). Instead, in the no SB43 organoids, with the most prolonged TGF-β signaling gradient, the apical perimeter and VZ thickness resulted akin to the ones of the neuroepithelium structures formed in the ENOs (Supplementary Fig. 5c–g). This again suggests that a

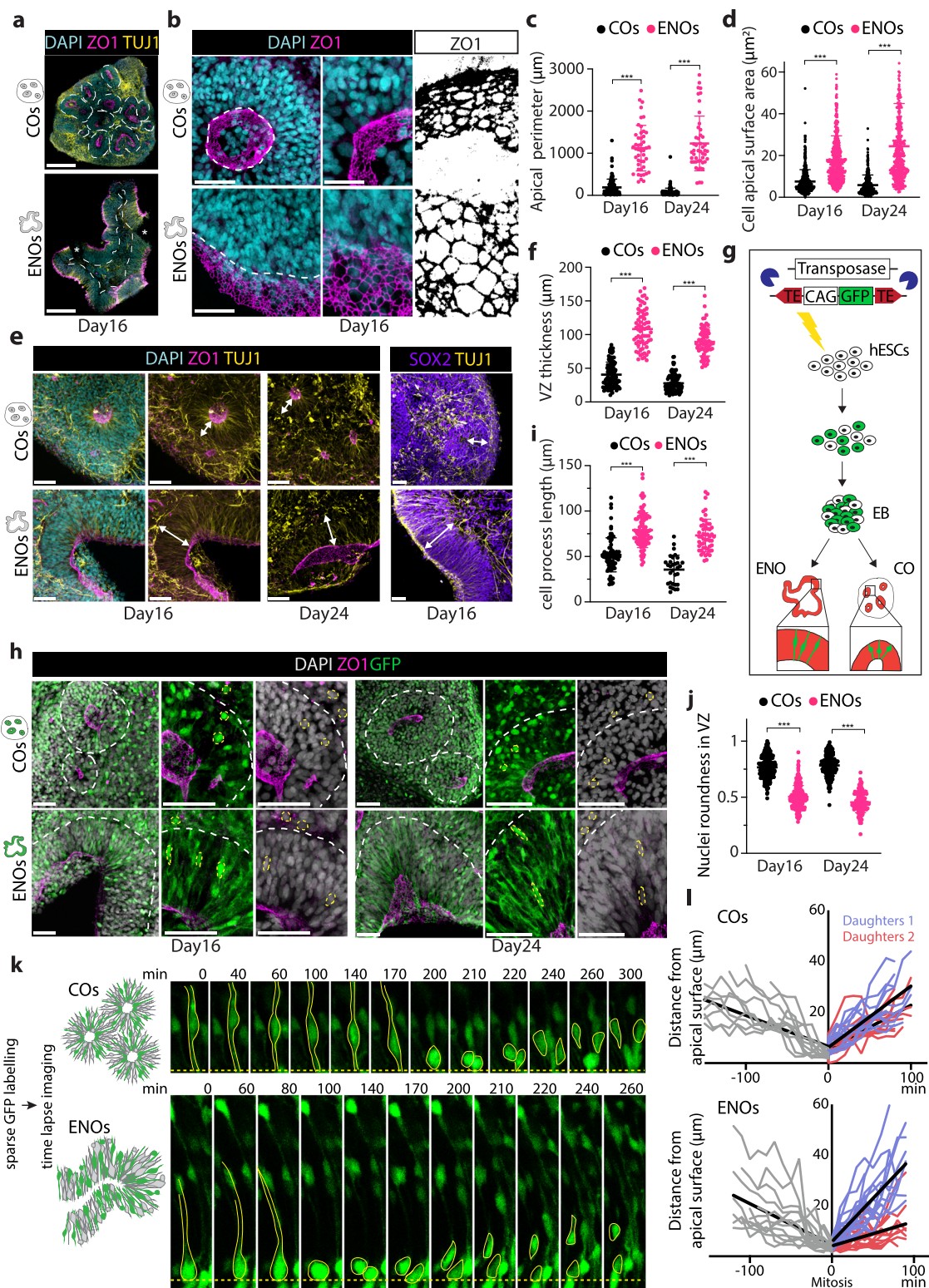

prolonged TGF-β signaling in a gradient fashion is important in driving these morphological changes.

To better assess cell morphology in the different organoids, we used a piggyBac transposable GFP system to generate sparsely labeled GFP+ hESC lines (Fig. 5g). Apical progenitor cells in the ENOs displayed a much more elongated shape, with longer apical-basal processes (Fig. 5h, i). Nuclei appeared more densely packed and presented with a more elongated shape in the VZ as compared to the nuclei of COs (and CommOs) (Fig. 5h, j and Supplementary Fig. 12g–k). Notably, a clearly

distinct area composed of elongated, dense nuclei with more round, sparse nuclei layer above the SOX2 + VZ was visible in the ENOs (Fig. 5h and Supplementary Fig. 12g–k), which altogether more closely resembles the organization observed in the VZ/SVZ germinal zones in human fetal brain tissue[55–58].

Interkinetic nuclear migration constitutes one of the mechanisms important for the formation of the pseudostratified neuroepithelium during cortical development[59–62] and it has been observed in brain organoids[20,33,34]. Concomitant to thickening of the VZ, apico-basal

**Fig. 5 | ENOs display tissue architecture features which more closely recapitulate human brain development. a** Representative immunofluorescence images of COs and ENOs stained for ZO1 and TUJ1. Dashed lines delineate basal side of rosettes/neuroepithelium structures. **b** Representative immunofluorescence images of COs and ENOs stained for ZO1. Dashed lines delineate apical side of rosettes/neuroepithelium structures. The black-and-white mask highlights differences in apical surface areas (right). **c** Quantification of the apical perimeter of rosettes/neuroepithelium structures of COs and ENOs. Dots represent individual rosettes/neuroepithelium structures. Mean ± SD is plotted. ***$p < 0.001$; Two-tailed unpaired $t$-test. **d** Quantification of cell apical surface area of cells in COs and ENOs. Dots represent cells. Mean ± SD is plotted. ***$p < 0.001$; Two-tailed unpaired $t$-test. **e** Representative immunofluorescence images of COs and ENOs stained for ZO-1 and TUJ1 (left) and SOX2 and TUJ1 (right). Arrows indicate the VZ. **f** Quantification of VZ thickness in COs and ENOs. Dots represent neuroepithelium/rosette structures. Mean ± SD is plotted. Sample size as in **c**. ***$p < 0.001$; Two-tailed unpaired $t$-test. **g** Schematic illustration of the experimental strategy to generate GFP-labeled ENOs and COs. **h** Representative immunofluorescence images of sparsely-labeled GFP+ COs and ENOs stained for ZO-1. White and yellow dashed lines delineate basal side of rosettes/neuroepithelium structures and nuclei shape, respectively. **i** Quantification of the cell process length in the VZ of ENOs and COs. Dots represent cells. Mean ± SD is plotted. ***$p < 0.001$; Two-tailed unpaired $t$-test. **j** Quantification of nuclei roundness of cells in the VZ of COs and ENOs. Dots represent nuclei. Mean ± SD is plotted. ***$p < 0.001$; Two-tailed unpaired $t$-test. **k** Representative still frames of live imaging of GFP-labeled cells undergoing mitosis at the apical side in day 14 ENOs and COs. **l** Comparative nuclear tracking of mitotic cells in COs and ENOs. Each line indicates the distance from the apical side over time before (gray line) and after mitosis (blue and red lines = daughter cells). Black lines indicate the mean. Scale bars for **a** = 250 μm, **b** low mag = 50 μm, high mag = 25 μm, **e** and **h** 50 μm. Images in **a**, **b**, **h**, and **k** are representative of $n = 3$ independent experiments. Exact sample size and exact $P$ values for **c**, **d**, **i**, **j**, and **l** are provided in Source Data. Source data are provided as Source data file.

nuclear movement tends to increase[61] and interspecies differences in VZ thickness and density have been linked to differences in interkinetic nuclear migration[63]. We performed live imaging of 14 day-old sparsely GFP-labeled organoids and analyzed individual cell movements before and after mitosis, which occurred at the apical side in both ENOs and COs (Fig. 5k). The nuclei of dividing cells in the ENOs tended to be closer to the apical side prior to mitosis, resulting in slightly reduced apicalwards movements as compared to the COs (Fig. 5k, l, Supplementary Fig. 13a, b). In the ENOs, one of the daughter cells displayed higher directionality and reached a different apicobasal nuclear position as compared to the other daughter cell (Fig. 5k, l, Supplementary Fig. 13a, b). In COs instead, the two daughter cells moved basalwards with similar directionality reaching a comparable apicobasal position (Fig. 5k, l, Supplementary Fig. 13a, b). This different division behavior between ENOs and COs was reproducibly observed across multiple cell divisions (Fig. 5l, Supplementary Fig. 13a, b). Interestingly, the mode of basalwards nucleokinesis observed in the COs is the least observed in vivo in mouse studies and it has been linked to periventricular nuclear overcrowding[61,64]. The increase in apico-basal nuclear movement observed in ENOs could possibly be linked to its thickened VZ and the associated dense cellular organization. Taken together, these observations suggest that ENOs are a valuable model to study neuroepithelium development.

## Discussion

Here, we show that sole introduction of temporal gradual neural induction majorly influences cortical organoid development, in terms of overall organoid architecture, cellular morphology, and importantly cortical fate specification. In particular, a simple temporal gradient during neural induction determines the formation of brain organoids composed of a single expanded apical-out neuroepithelium, instead of the uncontrollable appearance of multiple rosettes which each act as an independent center of neurodevelopment.

We find that prolonged exposure to a decreasing TGF-β/SMAD signaling gradient is important in this phenotype. TGF-β is necessary for the maintenance of stem cell pluripotency[45,46,65,66], which ultimately influences the formation of optimal telencephalic rosettes[67]. Dual SMAD inhibition is well-established to promote neural lineage specification in 2D culture[43] and has been extensively used in 3D brain organoid culture to direct cerebral cortex fate. The common theme across these existing protocols is the abrupt inhibition of signaling cues. Our findings show that TGF-β signaling dosage linked to timing is a critical factor influencing neural stem cell organization and better acquisition of dorsal identity in vitro. Interestingly, both *TGFB1* ligand and *TGFBR1-3* expression gradually decrease over time during the early stages of in vivo human brain development[68,69]. While we cannot fully exclude a role for the graded inhibition of BMP signaling (through DSM) in this process, the observation that ENOs generally formed with higher efficiency in the no SB43 condition (representing the condition in which the presence of TGF-β is the most prolonged during NI) as compared to the "conventional" ENOs protocol, corroborates the importance of prolonged TGF-β signaling. We additionally note some differences in ENO formation across different hESC lines linked to TGF-β inhibition strength. In fact, this also implies that generation of ENOs across different hESC/iPSC lines, beyond the here-tested H1, H9, and H14 lines, may benefit from initial evaluation of different concentrations, gradient steepness, and length of TGF-β inhibition.

Possibly, the more gradual neural induction as applied in the ENOs might provide the time for the induced neural stem cells to better organize into a single expanded neuroepithelium and to remain into a longer expansion phase. The macroscopic morphological changes observed in the ENOs come with a series of cytoarchitectural changes. In fact, increased VZ thickness, enlarged apical cell surface, and elongated nuclear as well as cellular shapes are critical features of the developing human brain that are also recapitulated by the ENOs. Since better cortical specification is also found in the ENOs, it is intriguing to speculate that tissue/cellular architecture and identity are two interconnected properties. While not as extensively characterized, the expanded neuroepithelium structures formed under no SB43 conditions likewise displayed enhanced cortical identity. Intriguingly, while the necessity of dual SMAD inhibition in 2D culture has been shown to improve cortical fate[43], the presence of SB43 appears superfluous in ENOs, perhaps related to its peculiar 3D architecture. Whether this more generally applies also to cortical organoids (and thus 3D culture) would be of interest to study in the future.

We noted a major impact on tissue polarity associated with ENO formation and the TGF-β gradient. While sudden NI forms rosette-containing cortical organoids, prolonging the presence of TGF-β during temporal-gradient NI (ENOs or no SB43 conditions), results in "apical-out" organoids, generating a single expanded neuroepithelium demarcated by continued ZO-1/NCAD positivity on the outside of the organoids. Of note, TGF-β and BMP signaling have been previously implicated in the initiation of invagination and closure of the neural tube during brain development[70,71]. Interestingly, the addition of exogenous ECM directly affects the formation of a single expanded neuroepithelium, which goes hand in hand with a change in polarity. In fact, ENOs retain apical-out polarity until ECM is provided during the maturation period (day 25 onwards), in which the neuroepithelium structures adopt a CO-like polarity. Further testing of ECM addition in different paradigms showed that ECM likely initiates tissue invagination in the ENOs. In this regard, exogenous ECM was recently shown to affect the timing of rosette polarization within telencephalic organoids[49]. Our observations on ENO formation, their polarity, and their overall tissue architecture, altogether suggests an intricate relationship between extracellular matrices and endogenous morphogen gradients in establishing polarity and expansion of the neuroepithelium.

Taking all these features in consideration, it emerges that ENOs and COs are drastically different at early stages. Several cell morphological, behavioral and identity features of the ENOs, resembling in vivo observations, make them potentially attractive as a model to study early human telencephalic development. These could include studying aspects of the formation of the neuroepithelium, as well as in-depth investigation of morphological/cytoarchitectural features e.g., related to developmental-associated disorders. Additionally, the presence of a single neuroepithelium instead of multiple rosette structures could help reproducibility and more homogenous cellular identity within and across different organoids. We could envision the particular single-structure organization of the ENOs as a future platform to implement cortical regionalization. The apical-out organization of the ENOs prior to addition of ECM on the one hand makes the stem cell-apical side more accessible, and therefore, it could allow easier targeting/manipulation of the neural stem cells[72]. On the other hand, it might pose more constrained space and access to nutrients for the developing of the neurons. However, addition of ECM after ENO structure formation determines an overall organoid polarity switch, akin to COs. Immunofluorescent analysis of a few key neuronal markers of ENOs and COs at later stages in culture suggested similar developmental cell composition, including lamination patterns and the presence of astrocytes. The possibility to further exploit the interplay between ECM and morphogen gradients could possibly help further optimization of the ENOs e.g., to translate the expanded neuroepithelium with enhanced cortical identity into improved later stages of development.

To conclude, the approach to generate ENOs is straightforward and easy to experimentally perform. The here-applied temporal gradient methodology also constitutes the starting point to explore additional gradients of different signaling pathways which may be important in shaping and defining different brain areas, as well as during later differentiation stages, to develop even further advanced human brain organoid models.

## Methods

### hESC culture
hESCs were used in accordance with the local ethical regulations. Stem cell lines were authenticated through STR profiling. Feeder-free human H1 (WA01), H9 (WA09), and H14 (WA14) hESCs (WiCell) were cultured on 6-well plates coated with hESC Qualified Matrigel (Corning, #354277) at 37 °C and 5% $CO_2$ in mTeSR Plus medium (Stem Cell Technologies, #100-0276). Culture medium was changed every other day and cells were passaged using Gentle Cell Dissociation Reagent (Stem Cell Technologies, #100-0485) every week, according to the manufacturer's protocol. Prior to passaging, cells were checked for regions of differentiation and if present, these were carefully scraped off with a pipette tip. Cultures were regularly tested for mycoplasma and tested negative without exception. The hESCs used throughout the study were maintained below passage 50.

### hESC gene editing
To generate GFP-labeled cells, H1 hESCs were electroporated with a transposon system using two plasmids (one encoding for the PiggyBac transposase and a transposable CAG-GFP plasmid), as follows, based on refs. 73,74. Prior to electroporation, confluent hESCs were detached from a Matrigel-coated well using Gentle Cell Dissociation Reagent, according to the manual. Detached colonies were mechanically dissociated into single cells and washed once with DPBS (Gibco, #14190-094). Half of a confluent well of a 6-well plate (roughly $1 \times 10^6$ H1 cells), were resuspended in 150 µl Opti-MEM (Gibco, #11058-021) containing 10 µg DNA mixture (3:1 CAG-GFP:PiggyBac) and incubated for 5 min at room temperature. The DNA-cell suspension was transferred to a 2 mm gap cuvette (Nepa Gene, #EC-002S) and electroporated with a NEPA21 electroporator using the following parameters for Poring Pulse: Voltage = 150 V, Pulse Length = 5 ms, Pulse Interval = 50 ms, Number of

Pulses = 2; and for Transfer Pulse: Voltage = 20 V, Pulse Length = 50 ms, Pulse Interval = 50 ms, Number of Pulses = 5. Immediately after electroporation, 1 ml of mTeSR medium containing 10 µM ROCK inhibitor Y-27632 (Abmole Bioscience, #M1817) was added to the cuvette and incubated at room temperature for 5 min. The cells were then washed once with mTeSR medium containing 10 µM Y-27632 and subsequently plated on a hESC Qualified Matrigel coated 12-well plate and cultured in mTeSR medium containing 10 µM Y-27632. After 24 h, medium was replaced with mTeSR without Y-27632. GFP-labeled hESC colonies were selected via manual picking. GFP-positive colonies were selected under a fluorescence microscope, detached by scraping and afterward cultured and maintained like wild type (WT) hESCs cultures.

### Generation of expanded neuroepithelium organoids (ENOs)
To generate expanded neuroepithelium organoids (ENOs), on day 0, confluent hESCs were detached from Matrigel-coated plate using Gentle Cell Dissociation Reagent and further mechanically dissociated into single cells. Cells were counted and resuspended in mTeSR containing 10 µM Y-27632 to reach a concentration of 9000 cells per 100 µl medium. Consequently, 9000 cells were added to each well of a 96-well ultra-low-attachment U-bottom plate (Corning, #10023683). The plate was spun down at 100 g for 1 min prior to incubation. After 48 h, on day 2, when embryoid bodies had formed, 100 µl of neural induction medium consisting of 1:1 DMEM/F12 (Life Technologies, #11330-032), 20% knockout serum replacement (Life Technologies, #10828028), 1X non-essential amino acids (Gibco, #11140-035), 1X GlutaMax (Gibco, #35050038), 1X β-mercaptoethanol (Gibco, #21985-023), and supplemented with 5 µM dorsomorphin (Sigma-Aldrich, #72102) and 10 µM SB-431542 (Stem Cell Technologies, #72232), was added to each well, thus creating a 1:1 ratio of mTSER:neural induction medium. On day 4 and 5, 50% of the medium was replaced by removing 100 µl and adding 100 µl of neural induction medium. On day 6, the medium was fully replaced by removing the full 200 µl and adding 200 µl of expansion medium consisting of Neurobasal-A (Life Technologies, #10888-022), 1X Penicillin and Streptomycin (Gibco, #15070-063), 1X B27 supplement without vitamin A (Life Technologies, #12587010), 1X GlutaMax (Gibco, #35050038) and supplemented with 20 ng/ml hEGF (Peprotech, #100-15) and 20 ng/ml hFGF2 (Peprotech, #100-18B). We noted that the efficiency of ENOs formation may vary slightly depending on the batch and stem cell line used. To improve ENO formation for a specific line, we occasionally employed a lower SB-431542 concentration (5 µM) in the neural induction medium, or, alternatively, we cultured EBs one day longer in mTeSR (till day 3) prior to starting the ENO protocol (as done in this study for H9 ENOs). On day 7, individual organoids were moved into expansion medium, each into a well of a 24-well plate treated with Anti-Adherence Rinsing Solution (Stem Cell Technologies, #07010) according to manufacturer's protocol and expansion medium was refreshed every other day, till day 24. On day 25, medium was fully switched to maturation medium consisting of Neurobasal-A (Life Technologies, #10888-022), 1X Penicillin and Streptomycin (Gibco, #15070-063), 1X B27 supplement with vitamin A (Life Technologies, #17504044), 1X GlutaMax (Gibco, #35050038) and supplemented with 20 ng/ml hBDNF (Peprotech, 450-02), 20 ng/ml NT-3 (Peprotech, #450-03) and 0.5% (v/v) Matrigel (Corning, #354234). Plates were moved to an orbital shaker (60 rpm) and medium was refreshed every other day till day 42. From day 43 onwards, organoids were maintained in maturation medium without BDNF and NT-3, with medium changes every 4 to 6 days. To assess ENOs formation efficiency, organoids were evaluated from day 14 onwards based on brightfield images analysis. At this stage the difference in morphology between COs and ENOs becomes apparent. ENOs formation is scored by the presence of convoluted and elongated structures, as opposed to COs, which instead display the typical spherical organoid shape. Of note, successfully formed ENOs are overall more translucent compared to COs when analysed with

brightfield imaging. The formation of a continuous neuroepithelium characteristic of ENOs was further confirmed by immunostaining. To generate ENOs with sparsely labeled GFP cells, GFP-labeled hESCs were mixed with WT hESCs at different ratios (5%, 10%, and 20%) to generate a mixed population. Seeding and culturing was identical to WT ENOs.

### Generation of cortical organoids (COs)

The generation of cortical organoids (COs) was performed as previously described[12,75], with slight modifications. Briefly, confluent hESCs were mechanically dissociated to single cells after detaching them from Matrigel-coated plates using Gentle Cell Dissociation Reagent. Then, 9000 cells per well were plated in a 96-well ultra-low-attachment U-bottom plate in 100 µl of mTeSR medium containing 10 µM Y-27632. On day 2, medium was fully replaced to the neural induction medium containing 1:1 DMEM/F12 (Life Technologies, #11330-032), 20% knockout serum replacement (Life Technologies, #10828028), 1X non-essential amino acids (Gibco, #11140-035), 1X GlutaMax (Gibco, #35050038), 1X β-mercaptoethanol (Gibco, #21985-023), and supplemented with 5 µM dorsomorphin (Sigma-Aldrich, #72102) and 10 µM SB-431542 (Stem Cell Technologies, #72232). Medium was fully refreshed on day 4 and day 5. On day 6, medium was changed to the expansion medium containing Neurobasal-A (Life Technologies, #10888-022), 1X Penicillin and Streptomycin (Gibco, #15070-063), 1X B27 supplement without vitamin A (Life Technologies, #12587010), 1X GlutaMax (Gibco, #35050038) and supplemented with 20 ng/ml EGF (Peprotech, #100-15) and 20 ng/ml FGF2 (Peprotech, #100-18B). On day 7, organoids were moved to a 24-well plate treated with Anti-Adherence Rinsing Solution as described in the product manual. Expansion medium was refreshed every other day, till day 24 and on day 25, medium was changed to maturation medium containing Neurobasal-A (Life Technologies, #10888-022), 1X Penicillin and Streptomycin (Gibco, #15070-063), 1X B27 supplement with vitamin A (Life Technologies, #17504044), 1X GlutaMax (Gibco, #35050038) and supplemented with 20 ng/ml BDNF (Peprotech, #450-02), 20 ng/ml NT-3 (Peprotech, #450-03) and 0.5% dissolved Matrigel (Corning, #354234). At this stage, plates were moved on a shaker (60 rpm) and medium was refreshed every other day till day 42. From day 43, organoids were maintained in maturation medium not containing BDNF and NT-3, with medium changes every 4 to 6 days. To generate COs with sparsely labeled GFP cells, GFP-labeled H1 hESCs were mixed with WT H1 hESCs at different ratios (5%, 10%, 20%) to generate a mixed population. Seeding and culturing of the organoids was identical to WT COs.

### Generation of commercial organoids (CommOs)

The STEMdiff Dorsal Forebrain Organoid Differentiation kit (Stem Cell Technologies, #08620) was used to generate Commercial Organoids (CommOs). The manufacturer's protocol was followed with slight modifications. In brief, hESCs were detached from the Matrigel-coated plate and dissociated into single cells as described above. Then, 9000 cells were plated in each well of a 96-well ultra-low-attachment U-bottom plate in Seeding Medium. On day 1, day 2, and day 5, 3/4 of the medium was removed from each well and substituted with the same amount of Forebrain Organoid Formation Medium. On day 6, formed aggregates were moved to a 6-well plated treated with Anti-Adherence Rinsing Solution in Forebrain Organoid Expansion Medium. A maximum of 30 aggregates were moved into a single well. To avoid fusion of the aggregates, plates were gently rocked back-and-forth and side-to-side to distribute the aggregates across the well prior to incubation. From day 8 till day 24, the medium was fully refreshed every 2 days. On day 25, medium was changed to Forebrain Organoid Differentiation Medium, and this was refreshed every 2 to 3 days. On day 43, medium was changed to Forebrain Organoid Maintenance Medium, and this was refreshed every 2 to 3 days for long-term maintenance.

### Perturbation of TGF-β signaling

The formation of ENOs relies on a stepwise gradient of TGF-β and FGF2, present in the mTeSR Plus medium[45,76], which is used for the first stages of ENO formation. Since the neural induction medium also contains a TGF-β inhibitor (SB-431542), we further evaluated the role of TGF-β through (1) omission of the TGF-β inhibitor or (2) providing stronger TGF-β inhibition, as follows. For (1), cells were plated as previously mentioned and on day 2, 100 µl of neural induction medium without SB-431542 was added. This was refreshed 50% on days 4 and 5, and from day 6 onwards the same protocol as for the ENOs was followed. For (2), to more strongly inhibit TGF-β, SB-431542 and dorsomorphin were added in a non-gradient fashion. Cells were plated as previously mentioned and on day 2, 100 µl of neural induction medium was added, supplemented with 10 µM dorsomorphin and 20 µM SB-431542, double the concentration normally used for the generation of the ENOs. From day 4 onwards, the same protocol as for the ENOs was followed.

### Matrigel addition at earlier timepoints during ENO formation

To assess if the addition of extracellular matrix would affect the development or characteristics of ENOs, Matrigel was dissolved in the expansion medium at different timepoints during the expansion phase of the ENO protocol. We tested different concentrations (0.05%, 0.2%, or 0.5% (v/v)) of Matrigel added to the expansion medium either at day 7, 10, or 18. For all conditions, Matrigel was thereafter added to the medium at every medium change. Organoids were moved to an orbital shaker at day 25 in all conditions. From day 25 onwards the ENO protocol was followed as described above.

### Organoid processing and immunofluorescence staining

Organoids were fixed at different timepoints in 10% buffered formalin overnight 4 °C on a rocker. Fixed organoids were washed 2–3 times in 1 x Phosphate Buffered Saline (PBS) at room temperature and stored at 4 °C prior processing. Samples were embedded in 3% UltraPure Low Melting Point Agarose (Thermo Fisher, #16520050) in PBS and sectioned to 40 µm using a Vibratome (Leica, #VT1200S). Sections were stored long-term in Cryo Preservation Solution (50% 2X PBS, 25% Ethylene Glycol and 25% Glycerol) at −20 °C. Sections were washed in PBS prior to immunostaining. Sections were mounted on Superfrost Plus Adhesion Microscope Slides (Epredia, #J1800AMNZ). Blocking and permeabilization were performed for 1 h at room temperature in 5% BSA (Sigma Aldrich, #A9418) and 0.2% Triton-X100 (Sigma Aldrich, #X100) diluted in PBS, in a humidified chamber. Primary antibodies were diluted in 2.5% BSA in PBS as listed in Supplementary Table 1. Primary antibodies were incubated for 3 nights at 4 °C in a humidified chamber. After washing 3 times for 10 min in PBS, sections were incubated overnight with the secondary antibodies, listed in Supplementary Table 1, diluted 1:1000 in 2.5% BSA in PBS at 4 °C in a humidified chamber. Sections were then incubated with DAPI diluted 1:1000 in PBS for 10 min at room temperature and further washed 3 times for 10 min with PBS. Sections were mounted using Immu-Mount mounting medium (Thermo Scientific, #9990402) and a glass coverslip (Epredia, #BB02400550AC13MNZ0). Whole organoid imaging was performed on unsliced organoids following the mLSR-3D protocol with minor changes[77]. Briefly, blocking and permeabilization of whole fixed organoids was performed for 4 h at 4 °C in washing buffer 1 (0.2% Tween-20, 0.02% SDS, 0.2% Triton X-100 and 0.2% BSA in PBS). Primary antibodies were diluted in washing buffer 2 (0.02% SDS, 0.1% Triton X-100 and 0.2% BSA in PBS) and incubated with whole organoid for 2 nights at 4 °C on orbital shaker. After washing 3 times for 1 h in washing buffer 2 at 4 °C on orbital shaker, secondaries antibodies were diluted in the same buffer and incubated for 2 nights, at 4 °C on orbital shaker. Organoids were further cleared using a FUnGI gradient (50% (v/v), 9.4% (v/v) dH2O, 1.1 mM EDTA, 10.6 mM Tris Buffer, 2.5 M Fructose, 2.5 M Urea) (33%, 66% and 100% (v/v) in washing buffer 2). Organoids were incubated for at least 1 h in each gradient step at 4 °C on orbital shaker.

Organoids were thereafter mounted in FUnGI between 2 coverslips, using silicone as sealant.

## Brightfield and confocal imaging

Brightfield images of organoids were acquired at different time points (days 7, 14, 20, 25) using an inverted microscope (Leica), with a 2.5x and 5x objective. Stained slides were imaged using a confocal Sp8 microscope (Leica), with a 20X dry objective or 25X water objective. Whole organoids were imaged using a STELLARIS confocal microscope (Leica), with a 20X multi-immersion objective, set to oil. For sliced organoids, image stacks of 5 µm z-spacing were acquired at a speed of 200 Hz and a 1024 × 1024 format, using 16-bit resolution. For whole organoid 3D imaging, image stacks of 1.56 µm z-spacing were acquired at a speed of 200 Hz and a 1024 × 1024 format, using 16-bit resolution. Files were exported in lif format for analysis.

## Image processing and analysis

Organoid morphological parameters were manually quantified for the different timepoints. To measure the organoid perimeter and the area of whole organoids, brightfield images were analyzed in ImageJ. Using the polygon selection tool, the outside boundaries of the organoid were rigorously outlined, and the organoid area and organoid perimeter were measured. The circularity of the organoids was measured on the same brightfield images using the "shape descriptors" plugin present in ImageJ. Merged Lif files of immunofluorescence images were processed and analysed using ImageJ and Photoshop. To quantify SOX2 positive cells over whole organoids, up to 12 randomized areas were selected within the germinal zone of each organoid and nuclei positive for SOX2 were manually counted. Numbers of SOX2+ cells were normalized over DAPI. To quantify DCX abundancy, the area occupied by the positive fluorescence DCX signal was quantified in and above the germinal zones, which was then normalized over the area of the organoid. For the quantification of rosette PAX6 positivity, rosettes were scored either as positive (+) when PAX6+ cells were homogenously distributed, negative (−), when virtually no PAX6+ cells were observed, or mixed (+/−) when a rosette showed both PAX6 positive and negative portions. The distribution was represented as percentage over the total amount of quantified rosettes. The same assignment was done for EMX2 and EMX1 staining. To quantify PAX6 and EMX2 positive cells in VZ areas, nuclei positive for the markers were manually counted in three unbiasedly selected areas of each rosette (in the COs) or neuroepithelium structures (in the ENOs). For each quantified organoid, three or more rosettes (in the COs) or neuroepithelium structures (in the ENOs) were selected for counting. Numbers of PAX6+ and EMX2+ were normalized over DAPI. The percentage of VZ areas per organoid was measured based on NCAD staining and normalized over the total area of the organoid. The apical perimeter of the rosette (in the COs) or neuroepithelium structures (in the ENOs) was determined by manually delineating the perimeter of the apical side of each rosette/neuroepithelium structure, based on NCAD staining. The thickness of rosettes (in the COs) or neuroepithelium structures (in the ENOs) was measured based on SOX2 positivity, until the appearance of the TUJ1+. Thickness was measured by averaging the length from the apical to basal side in three different and randomized areas for each rosette/neuroepithelium structure. For both apical perimeter and VZ thickness, multiple rosettes/ neuroepithelium structures were measured per organoid. Quantification of the apical surface area of individual neural progenitor cells was performed by manually delineating ZO-1 positive cell perimeters on the apical side of rosettes/neuroepithelium structures. A minimum of 20 cells were quantified for each organoid. Nuclei roundness was measured using the "shape descriptors" plugin in ImageJ, by delineating the perimeter of DAPI positive nuclei. Roundness is measured by the following formula: 4*area/pi*sqr(major axis). Nuclei were considered part of the VZ if they were present in rosettes/neuroepithelium structures delineated by NCAD+, SOX2+ and TUJ1−. Nuclei just above were considered

as part of the above VZ. Cell processes were measured in organoids generated using sparsely GFP-labeled ESCs using the segmented line tool on ImageJ. 3D reconstruction videos of whole organoids were generated using Imaris software.

## Live imaging and analysis of cell division in GFP-labeled organoids

Sparsely labeled GFP COs and ENOs were imaged in the expansion phase between day 13 and day 16 using a confocal Sp8 microscope (Leica), with a 10X dry objective. Prior to live-imaging, whole organoids were individually placed in a well of a flat-bottom 96-well plate. Organoids were imaged under environmentally controlled conditions at 37 °C and 5% $CO_2$ using a confocal Sp8 microscope with a motorized stage. Image stacks were acquired using a 488 nm laser every 10 min for 4 to 12 h with 6 µm z-spacing at 16-bit resolution. The analysis of interkinetic nuclear migration was performed by measuring the distance (using ImageJ) from the apical surface of the moving cell body (as an approximation for the nucleus) of the dividing cell in each temporal frame towards the apical side (apicalward movement) and, after mitosis, of the daughter cells from the apical side upwards (basalward movement).

## RNA extraction and qPCR analysis

To isolate RNA, individual organoids were collected in 500 µl TRIzol Reagent (Thermo Fisher, #15596026). Samples were fully lysed by repetitive pipetting and then snap-frozen in liquid nitrogen. RNA was extracted using isopropanol precipitation using the TRIzol manufacturer's protocol and resuspended in mQ sterile water. RNA concentrations and purity were determined using a NanoDrop spectrophotometer. Then, 250 ng of RNA was used for cDNA synthesis using the SuperScript IV kit (Thermo Fisher, #18091050). The cDNA reaction was diluted 1:10, and 2 µl of the diluted cDNA was used for each qPCR reaction. qPCR reactions were performed using the iQSYBR Green mix (Bio-rad, #1708887). The qPCR primers are listed in Supplementary Table 2.

## Statistical analysis and data visualization

All experiments were performed with multiple samples across multiple independent organoid batches as detailed in the figure legends and in Supplementary Table 3. Statistics were performed and graphs were generated using GraphPad Prism (version 9.1.2). Two-tailed unpaired t-tests and nonparametric Mann–Whitney $U$ tests were performed as statistical analysis and standard deviation was used as measure to show data dispersion.

## Statistics and reproducibility

The specific sample size and $P$ values for all performed analyses are reported in the Source Data file if not stated in the Figure Legends. Each organoid batch is defined as an independent experiment.

## Reporting summary

Further information on research design is available in the Nature Portfolio Reporting Summary linked to this article.

## Data availability

All data generated in this study are provided in the Supplementary information files or in the Source Data file, or from the corresponding authors upon reasonable request. Source data are provided with this paper.

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

## Acknowledgements

We thank Dr. Marcel Kool and Dr. Jens Bunt for sharing of resources, Dr. Ravian van Ineveld and the PMC imaging center for 3D imaging support, Dr. Simone Massalini for technical support, and Francesco Andreatta for helpful discussions. Part of this work is supported by an Open Competition Science-M grant from the Dutch Research Council (NWO) to B.A. and D.H. is supported by a NWO VENI grant (VI.Veni.212.134).

## Author contributions

Conceptualization, supervision, project administration, funding acquisition, writing—review & editing: D.H. and B.A.; Methodology, writing—original draft, and visualization: A.P., D.H., and B.A.; Formal analysis and data curation: A.P., R.F., D.H., and B.A.; Investigation: A.P., R.F., I.Z., and S.B.; Resources: H.C., D.H., and B.A.

## Competing interests

D.H. and B.A. are inventors of patents related to organoid technology. H.C. is inventor of several patents related to organoid technology; his full disclosure is given at https://www.uu.nl/staff/JCClevers/. The other authors declare no competing interests.
