## [Peer Review File · Nature Communications]

Temporal morphogen gradient-driven neural induction shapes single expanded neuroepithelium brain organoids with enhanced cortical identityREVIEWER COMMENTS

Reviewer #1 (Remarks to the Author):

In this manuscript, the authors assess the effects of using a graded transition of morphogens and small molecule inhibitors during neural patterning in embryoid bodies to generate brain organoids. Surprisingly, the authors find that the organoids contain more-or-less one continuous neuroepithelium with dramatically different shape and organizational properties than traditional cortical organoids. They also find more efficient patterning to dorsal cortical (PAX6/EMX2+) identity. The overall approach and findings are novel and provide an important avenue for further developing brain organoid models to better recapitulate aspects of in vivo human cortical development. However, this new protocol seems to be at a preliminary stage of development with all the work being done on one hESC line and limited measures of consistency shown, and there are other important issues that the authors should address to put the work in a proper context.

Concerns:

- 1) The use of “Morphogen Gradient” in the title is somewhat misleading. To most readers this would indicate a spatial morphogen gradient that is important for developmental axes. However, the authors seem to mean a temporal gradient, and – more to the point – they mean a stepwise transition. The 4, 50% media changes are not a true “gradient” but may be better described as a graded transition. The wording for the title and text should be changed.
- 2) The entire study seems to have been done with the H1 hESC line. The authors need to show that their protocol works for multiple hESC/iPSC lines.
- 3) How consistent are the ENOs from batch to batch? The use of qRT-PCR, scRNA-seq or more comprehensive marker immunostaining on multiple organoids from multiple batches are needed to show consistency.
- 4) Do the ENOs have an “inside-out” polarity? Several of the images suggest that this is the

case:

- a. N-cadherin, radial TUBB3, dividing KI-67+ cells are all seen on the outside of the ENOs (Figs. 1e, 2e, 3f; Supp Fig. 2). These are also seen on the inside of the rosettes on the apical lumen for COs (Figs. 1e, 2e, 3f, and prior work).
 - b. Based on the full ENO images in Supp Fig. 2, any empty space is primarily on the outside of the ENOs rather than a central lumen, and NCAD and SOX2 appear to be located on the outside of the ENOs.
 - c. Do the panels with typical concave “lumen”/VZ layers (e.g., Figs. 1e and 4d) actually represent the invaginations seen on the outside of the entire ENOs? If true, the “luminal perimeter” would actually be measuring the basal side of the neuroepithelial structure, which in brain organoid structures cannot be called a “lumen.” This potential inside-out inversion could cause the drastic differences in apical surface areas and nuclei roundness due to geometric constraints.
 - d. Whole ENO images of clear apical markers such as ZO1 or PKCz would help definitively establish the polarity for the reader.
 - e. Other groups (Watababe et al, 2022) utilize TGF-beta to generate “inside-out” organoids (NCAD and PKCz on the exterior) at similar timepoints. However, at later time-points (8 weeks) the continuous outside neuroepithelium “pinches off” into many rosettes. It would be helpful to see full structures of later ENOs to see if they generate multiple lumen/rosette structures or maintain a single structure.
- 5) Some replicate group sizes were very low (3-4 organoids). At this low level, it does not matter that 3 independent experiments are represented because it is not possible to get an accurate idea of the variance of a group or between independent experiments. Other quantifications used multiple rosettes from the same organoid. Some color coding to see replicates from the same organoid or same distinction in the graph or statistics would be helpful to add.
- 6) The authors should describe the apical-basal polarity for Fig. 3m so that the reader can interpret the neuronal layer marker immunostaining. Also, using a layer 1 marker such as Reelin would be useful.

7) While the imaging with PAX6 and EMX2 are somewhat convincing, it would be helpful to have further evidence that this change in patterning is resulting in the same cell types as the CO methodology. Additional ICC markers, qRT-PCR, or scRNA-seq would be helpful in this regard.

8) It is unusual that oRG would be present in 16-day-old organoids, especially at the luminal surface. The staining in Fig. 3e is unusual and does not appear to be specific for oRG.

9) The TBR2 immunolabeling in Fig. 3D seems very diffuse with more in the VZ than SVZ. Lower magnification views should be shown.

10) Assessing ENOs at later timepoints to more clearly define cortical lamination patterns, the formation of astrocytes, etc. would make the work more useful.

11) In terms of statistics, some data are obviously not a normal distribution (Fig. 3g, j) but a t-test was used. Although the significance is obvious by eye, please use a nonparametric test for these types of distributions.

12) The timelapse cell division data are not convincing with an N of 1 for each group. To truly know what type of cell division is more common, multiple examples or statistical analysis should be performed to provide better evidence. At the very least, multiple examples should be shown (some could go into the supplemental figures).

Reviewer #2 (Remarks to the Author):

Pagliari and colleagues develop a method to derive cortical organoids from PSCs with extended neuroepithelium (ENOs) as potential alternative to the currently derived cortical organoids (COs) containing multiple rosettes. The authors propose that such extended neuroepithelial structures mimic more faithfully early human embryonic cortical development, which favors germinal zone expansion, both laterally (length) and vertically (thickness) prior to transition to neurogenic mode.

The authors apply the dual Smad inhibition method as basis for their experiments to dissect factors governing epithelial cell expansion and find that a gradual switch from “full” TGF activation (pluripotent state) to “full” TGF inhibition (neural induction phase) enables lateral progenitor cell expansion on the expense of multiple smaller size rosettes. Authors show decreased neuronal marker expression in ENOs and suggest that by allowing a permissive zone of natural rather than abrupt decrease in TGF signalling (and in the presence of BMP inhibition), extended proliferation phase and delayed neurogenesis occur.

The authors assess cortical identity of ENOs using a number of markers expressed in the cortical epithelium including conventional stem cell markers, positional markers and oRG markers. They show that ENOs are more packed with cells expressing these stem cell and cortical markers when compared to COs – as sign for enhanced cortical specification and delayed neurogenesis.

The authors further show that ENOs exhibit larger ventricular regions delineated by area of luminal perimeter as well as thickness of germinal zone in rosettes / vesicles, and that these measures expand over time, while rosettes in COs decline in size with development. They finally show that apical progenitors bear longer apical-basal processes accompanied by a more densely packed nuclei performing interkinetic nuclear migration in a fashion reminiscent to that occurring in human cortical development in vivo.

This a very interesting study with great potential to the field for a number of reasons. First, a small / simple change made in the protocol makes a big difference in organoid outcome. Second, the change can be readily integrated into other protocols. Third, retaining rosette structure for longer periods is important in attempts to mimic early expansion phases of cortical development. Finally, the notion of a single expanded rosette as the in vitro counterpart of a single cerebral vesicle will allow future arealization studies within organoids.

There are a number of major requests that need to be addressed to strengthen the claims before the work can be considered for publication.

1. Enhanced cortical identity: the authors claim organoids are cortical and show PAX6, EMX2 as cortical and SOX2 as stem cell markers – but they do not show these markers together co-localized, nor they show at all expression of FOXG1. They should specifically show FOXG1 co-expression with these markers (a. FOXG1/SOX2 for telencephalic stem/progenitors, b. FOXG1/PAX6, FOXG1/EMX2 (or preferably FOXG1/EMX1) for dorsal telencephalic identity. PAX6/EMX1 is also a good combination. These findings should be then compared between COs and ENOs to demonstrate the enhanced cortical identity claimed.

2. Increased thickness of germinal zones: similarly, it is important to demonstrate that areas compared between ENOs and COs are composed of cortical stem cells (SOX2) or IP cells (TBR2) based on the above co-expression analyses.

3. The analyses are also required to be demonstrated in more lines (at least one more) - an iPSC line, so that less characterized cells can be tested for robustness, particularly with the dynamic range of endogenous TGF activation level across lines.

4. The authors should show more complete organoids in their staining (such as Supp Fig. 2) and overall organoid size analysis in ENOs vs COs.

While there is no absolute requirement to generate scRNA-Seq datasets for ENOs vs. COs, it is strongly recommended that authors perform such an analysis. This will provide a significant support to the finding that is currently missing due to the use of stainings only in one line only.

Other more minor points:

Fig 1a (scheme) is confusing. Per each treatment box all factors should appear together

under one color. Otherwise, it feels that each row is a different treatment (i.e. FGF2 is different treatment than FGF2).

Fig. 3a. Do Day 85 ENOs have any rosettes at all? The image on the right shows only scattered SOX2 cells but no rosettes. That would mean that there is no ventricular lining in late-stage ENOs. Larger area sections of early and late-stage ENOs through time can clarify this.

Fig. 3c. DCX (and SOX2 as well) should be counted within germinal zone only.

Fig. 3m: Can authors provide representative staining for COs to assess whether cortical lamination is more efficient in ENOs? If latter is correct then this will support an idea of better cortical diversity in ENOs.

Fig. 4a bottom left – not clear why ZO-1 faces out. Larger area images should help clarify this.

Reviewer #3 (Remarks to the Author):

Neural organoids have become a more widely used tool in recent years for modelling human brain development and disease. However, while directed differentiation approaches lead to more homogeneous cortical tissues, tissue architecture is often compromised with somewhat simplistic neural rosettes forming, rather than the elongated, dense telencephalic tissue of the in vivo brain. To overcome this, Pagliaro et al. have devised an interesting and simple method to generate more elongated neuroepithelium, that also reliably forms cortical identity. This is a careful and detailed study demonstrating an important methodological advance that will be vital to the field. I have only a couple comments, as detailed below:

1. The only major comment relates to the polarity of the tissues. Because there is no extracellular matrix provided, such as Matrigel as used in many other brain organoid

protocols, the ENOs maintain their apical surface on the external surface of the organoid. This naturally limits the expansion potential upon neurogenesis, as neurons can only go inside. It is also inside-out, compared with the in vivo brain. Finally, it also means the curvature of the neuroepithelium is not as it would be in vivo, since the geometry forces convex regions of curvature, whereas cortical neuroepithelium should have a concave curvature. While parts of the ENOs exhibit this curvature spontaneously (which is very interesting), many regions must instead fold outward. Thus, I wonder if the authors have tried adding Matrigel, either by embedding the ENOs at some stage after initial patterning, or by dissolving Matrigel in the media? This would be a very interesting further direction.

2. It would be important that the authors specify right at the beginning (i.e. in the title and abstract) that the gradient that they are talking about is temporal. As it stands, the wording could lead to initial confusion on the part of the reader, since in development gradients are usually spatial or spatio-temporal.

We have found the Reviewers' remarks constructive and revised our manuscript accordingly. In our revised version, we have added a significant number of new experiments and data.

To summarize the main revisions:

- 1) As suggested by Reviewers 1 and 2, we now show the reproducibility and robustness of ENO formation in multiple different hESC lines. Our results confirm that the temporal neural induction gradient induces ENO formation in all tested lines (H1, H9 and H14 hESC lines). The new data are reported in Figures 1, 2, and Supplementary Figures 1-4, 10.*
- 2) We considerably increased the 'n', both in terms of generated and analyzed batches and in the number of organoids analyzed per batch and performed key experiments in the different hESC lines. These data are now included in all the quantification panels in Figure 1-5 and Supplementary Figures 1-12.*
- 3) We provide a more robust characterization of the enhanced cortical identity in ENOs by increasing the number and type of stainings. Specifically, we confirmed the forebrain identity by FOXG1/SOX2 co-staining and the dorsal telencephalic identity by co-staining for EMX1, EMX2 and PAX6. We have additionally increased the molecular characterization of the ENOs by qPCR for several markers, which consistently show increased expression in the ENOs as compared to the COs (see Figure 4 and Supplementary Figures 2d, 8-10).*
- 4) We have now better clarified the polarity of the ENOs (amongst others by including 3D reconstruction videos (Supplementary Video 1-2)), and, as suggested by Reviewer 3, we have tested the effects of Matrigel addition at different concentrations/timepoints and showcased the effects on the polarity and structure of the ENOs. These data are included in Figure 5a-b and Supplementary Figure 6c-d, 7.*
- 5) Finally, we have addressed all the remaining points, e.g. by showing multiple low-magnification images of whole organoids (see e.g. Figures 1f, 4e, 4i, 5a and Supplementary Figures 3, 5b, 10, 11a) better and multiple example of various stainings (see e.g. Figure 4 and Supplementary Figure 8), more precise cellular definition of the germinal zones, and clarified the fate of the COs and the ENOs at later time points. We as well made several textual changes, including the suggested adjustments to the title.*

Below, we give a point-by-point rebuttal to the specific points. We have highlighted in our manuscript in red the more substantial changes.

Reviewer #1 (Remarks to the Author):

In this manuscript, the authors assess the effects of using a graded transition of morphogens and small molecule inhibitors during neural patterning in embryoid bodies to generate brain organoids. Surprisingly, the authors find that the organoids contain more-or-less one continuous neuroepithelium with dramatically different shape and organizational properties than traditional cortical organoids. They also find more efficient patterning to dorsal cortical (PAX6/EMX2+) identity. The overall approach and findings are novel and provide an important avenue for further developing brain organoid models to better recapitulate aspects of in vivo human cortical development. However, this new protocol seems to be at a preliminary stage of development with all the work being done on one hESC line and limited measures of consistency shown, and there are other important issues that the authors should address to put the work in a proper context.

A: We highly appreciate the positive evaluation of the Reviewer regarding the novelty and the importance of our study. We agree with his/her comments about the necessity to substantiate our study in additional lines and to provide a better characterization of efficiency and consistency. We have followed the Reviewer's suggestions and extensively revised the manuscript accordingly. In summary, we now show that ENOs can be robustly generated using different stem cell lines. Furthermore, we consistently increased the number of analyses and included different parameters to show consistency of the protocol. We also performed more in-depth characterization regarding the ENO's enhanced cortical identity. A detailed reply to the specific points is provided below.

Concerns:

1) The use of "Morphogen Gradient" in the title is somewhat misleading. To most readers this would indicate a spatial morphogen gradient that is important for developmental axes. However, the authors seem to mean a temporal gradient, and – more to the point – they mean a stepwise transition. The 4, 50% media changes are not a true "gradient" but may be better described as a graded transition. The wording for the title and text should be changed.

*A: Well taken. We indeed meant a temporal gradient. While the word "gradient" can refer to both spatial as well as temporal gradient, we agree to avoid unnecessary confusion, and we now changed the manuscript's title to **"Temporal morphogen gradient-driven neural induction shapes single expanded neuroepithelium organoids with enhanced cortical identity"**. Additionally, we now explicitly explain throughout the text that our protocol indeed relies on a stepwise transition, which we intended as an approximation of temporal gradient.*

2) The entire study seems to have been done with the H1 hESC line. The authors need to show that their protocol works for multiple hESC/iPSC lines.

A: The Reviewer is correct. In addition to the H1 line, we have now used two additional hESC lines (H14 and H9), collectively representing widely used lines for the generation of brain organoids (see e.g. Lancaster et al. Nature 2013, Kelava et al. Nature, 2022; Ogawa et al. Cell Rep 2018; Yao et al. Cell Stem Cell 2017; Esk et al. Science, 2020). We confirm that ENOs can be successfully generated using all 3 lines. We also observed some variation in ENO efficiency

formation across the different lines (i.e. a mean ENO efficiency formation of 64%, 100% and 45% across multiple experiments in the H1, H14 and H9, respectively; see Figures 1-2 and Supplementary Figures 1,2, and 4). Importantly, we also observed similar trends of responsiveness to TGF β signalling (Figure 2 and Supplementary Figure 4) and ENOs formed as well from H1, H9 and H14 in the no-SB43 condition (representing the condition in which TGF β is for the longest time present), with even higher efficiency. We have also included a section in the discussion elaborating on the line variability and TGF β dosage.

3) How consistent are the ENOs from batch to batch? The use of qRT-PCR, scRNA-seq or more comprehensive marker immunostaining on multiple organoids from multiple batches are needed to show consistency.

A: Following this suggestion, we have further analyzed the robustness and the consistency of ENOs formation at multiple levels.

First, we have generated multiple additional batches from the H1, the H14 and the H9 lines. The efficiency of ENOs formation has been compared across lines. As expected, we observed some experimental variability across the different conditions, but an overall high percentage of ENO formation in the various experiments (averaged 64%, 100% and 45% of ENOs formation across experiments in the H1, H14 and H9, respectively). We have now also highlighted ENO formation efficiencies within the different batches of these lines, as illustrated in Figure 1h and 2c. We have also included many more brightfield images and examples of ENOs formed from different hESC lines and batches. In addition, we have extensively revised the quantification of organoid characteristics (circularity, perimeter, area) across these different lines and displayed batch-to-batch differences. These new data are included in Figure 1d, 1e, 1h and 2c and supplementary figures 2 and 4.

The above measures validated that the formed ENOs are reproducible across batches. Next, we have further validated this by immunostainings using the markers originally presented in the manuscript, as well as extended our initial panel by adding novel stainings, such as EMX1 and FOXG1. Please note that the we have considerably increased the amount of analyzed organoids, including now characterization of multiple organoids across multiple batches, which is now highlighted in the figures when possible, or in the figure legends. These new data have been included in Figure 4 and supplementary figures 3, 8-10.

Finally, we now included extensive analysis of ENOs from different batches using qRT-PCR (Figure 4l), including both general and cortical-specific markers, previously used to demonstrate acquisition of cortical identity in brain organoids (see e.g. Rosebrock et al. Nature Cell Biology, 2022). These results corroborate the increased and better cortical specification in the ENOs as compared to the COs, as well as a good level of reproducibility across ENOs. We have also included expression data clarifying the ENO-to-ENO variation in expression of these markers (supplementary figure 2d). Taken altogether, these novel data support the robustness and reliability of ENO formation and their characteristics.

4) Do the ENOs have an “inside-out” polarity? Several of the images suggest that this is the case:

a. N-cadherin, radial TUBB3, dividing KI-67+ cells are all seen on the outside of the ENOs (Figs. 1e, 2e, 3f; Supp Fig. 2). These are also seen on the inside of the rosettes on the apical lumen for COs (Figs. 1e, 2e, 3f, and prior work).

b. Based on the full ENO images in Supp Fig. 2, any empty space is primarily on the outside of the ENOs rather than a central lumen, and NCAD and SOX2 appear to be located on the outside of the ENOs.

A: The Reviewer is correct. The ENOs indeed present their apical side on the “outside” of the ENOs, where indeed the NCAD and SOX2+ cells are located. Figure 5a and Supplementary Figures 7a and 11a highlight the ZO-1 marking the “outside” of whole-organoid ENOs. This is now better clarified in the text and we also added low-magnification images of whole ENOs and COs marked for ZO-1 to make this concept and difference between the polarity between ENOs and COs clearer. Finally, we have added movies in which we performed 3D reconstruction imaging of whole ENOs (see Supplementary videos 1-2), where we believe it is easier to visualize the complex 3D structure (which includes ZO-1 staining).

c. Do the panels with typical concave “lumen”/VZ layers (e.g., Figs. 1e and 4d) actually represent the invaginations seen on the outside of the entire ENOs? If true, the “luminal perimeter” would actually be measuring the basal side of the neuroepithelial structure, which in brain organoid structures cannot be called a “lumen.” This potential inside-out inversion could cause the drastic differences in apical surface areas and nuclei roundness due to geometric constraints.

A: We have defined the VZ and the “lumen” based on the apical side of the neuroepithelium, where indeed the SOX2, N-CAD, MKI67 and positive staining for the apical marker ZO-1 is located. This is also consistently where the apical progenitors undergo mitosis (see Figure 5k, Supplementary Figure 12). The generated neurons are present at the opposite, basal side (“inside” part). So, although the apical side is on the “outside” of the ENOs, “biologically” presents all the features of the neuroepithelium apical side, for this reason we referred to this as the lumen and luminal perimeter. However, we do agree that indeed a “closed lumen” is not present in the ENOs, and therefore we have changed the term “luminal perimeter” in the text with “apical perimeter” instead (Figure 5c, Supplementary Figure 11c). We have now added clarifications in the images and in the legends. We also now clarified what the white outlines indicate in the different panels.

We indeed agree that this change in polarity could cause the drastic differences in architectures. In fact, we show that this inside-out morphology is associated with differences in the apical surface area and nuclei roundness compared to what is observed in the conventional rosettes formed in the COs (Figure 5d, 5j and Supplementary Figure 11h-k). We have added a section in the discussion regarding the polarity of the ENOs.

d. Whole ENO images of clear apical markers such as ZO1 or PKCz would help definitively establish the polarity for the reader.

A: Agreed. As mentioned before, we have now added images of the apical marker ZO1 in whole ENOs (Figure 5a and Supplementary Figures 7a and 11a), which clearly highlights their different polarity as compared to COs. Finally, we have added movies in which we performed

3D reconstruction imaging of whole ENOs (see Supplementary videos 1-2), where we believe it is easier to visualize the complex 3D structure of the ENOs.

e. Other groups (Watababe et al, 2022) utilize TGF-beta to generate “inside-out” organoids (NCAD and PKCz on the exterior) at similar timepoints. However, at later time-points (8 weeks) the continuous outside neuroepithelium “pinches off” into many rosettes. It would be helpful to see full structures of later ENOs to see if they generate multiple lumen/rosette structures or maintain a single structure.

A: Indeed, as noted by the Reviewer, until around day 24, ENOs have an “inside-out” polarity. At later time points (day 24 onwards), organoids are switched to maturation medium (containing Matrigel) and moved onto an orbital shaker. We have now evaluated the ENOs shortly after this event (day 35), and show that the neuroepithelium continuum rearranges its architecture into multiple structures, and showing rather large, concave internal lumina, some of which resemble rosettes (Supplementary Figure 6b). As maturation and neurogenesis proceeds, small germinal-like structures are left amongst the abundant neurons, very similar to the ones observed in the COs. These later timepoints and lower magnification images of the ENOs are now shown in Figure 4f and Supplementary Figures 6b-d and 7. Please see also our detailed reply to the main comment of Reviewer 3, concerning the effect of Matrigel on the ENO polarity and architecture.

5) Some replicate group sizes were very low (3-4 organoids). At this low level, it does not matter that 3 independent experiments are represented because it is not possible to get an accurate idea of the variance of a group or between independent experiments. Other quantifications used multiple rosettes from the same organoid. Some color coding to see replicates from the same organoid or same distinction in the graph or statistics would be helpful to add.

A: Agreed. We now consistently increased the ‘n’ of the analysis shown throughout the paper (the figure legends are changed accordingly). We have considerably increased the number of batches analyzed (a minimum of 3 batches per each quantification), increased the amount of organoids within each batch (a minimum of 3 organoids per batch have been analyzed), and extended our findings by including two additional hESC lines on top of the H1 line (H14 and H19, as mentioned earlier). Whenever possible, we have highlighted and distinguished the results derived from the different batches and provided side-by-side comparisons of quantifications across batches (e.g. see Supplementary Figures 2 and 9). Altogether, these data reinforce the reliability of the ENO protocol and ENO characteristics. For simplicity, we now also provide a supplementary table elaborating on the number of organoids and number of batches used per line (Supplementary Table 3).

6) The authors should describe the apical-basal polarity for Fig. 3m so that the reader can interpret the neuronal layer marker immunostaining. Also, using a layer 1 marker such as Reelin would be useful.

A: Thanks for the suggestion. We have now indeed indicated the apical and basal side in the new Figure 3h (including former 3m).

7) While the imaging with PAX6 and EMX2 are somewhat convincing, it would be helpful to have further evidence that this change in patterning is resulting in the same cell types as the CO methodology. Additional ICC markers, qRT-PCR, or scRNA-seq would be helpful in this regard.

A: Agreed. To further show that the change in patterning in the ENOs results in the same cell types as the COs, we have now increased our panel of ICC markers and also performed qRT-PCR analysis for different brain- and cortical specific markers, as well as “off-lineage” markers (see Figure 4l and Supplementary Figures 1h and 2d). These results collectively confirm the forebrain identity of the ENOs (SOX2+FOXG1+) and the increased cortical specification as compared to the COs (based on comparative EMX1, EMX2, PAX6 co-stainings). In addition, similar to COs, ENOs do not display “off-lineage” markers, as shown by the lack of expression of mesodermal marker Brachyury and near-absent epithelial marker E-CAD, and endodermal marker SOX17. Instead, ENOs (like COs) robustly express the neuroepithelial marker N-CAD (Supplementary Figures 1h and 2d).

Of note, the initial change in specification (i.e. increased cortical identity) does not change the capacity to undergo neurogenesis at later timepoints, where we find similar neuronal cell types and layers are formed as compared to COs (see Figure 3h).

8) It is unusual that oRG would be present in 16-day-old organoids, especially at the luminal surface. The staining in Fig. 3e is unusual and does not appear to be specific for oRG.

A: Thanks for pointing this out. We agree that the PTPRZ1 staining in former Fig. 3e was not optimal. PTPRZ1, especially in early development, also weakly stains apical radial glia, at least in brain organoids (see for instance Rosebrock et al. Nature Cell Biology, 2022, Fig. 5e), which is presumably indeed reflected in the staining of our ENOs in previous Fig. 3e. In order to avoid confusion, we have removed this panel, as we indeed did not intend to robustly claim the presence of oRGs at this timepoint.

9) The TBR2 immunolabeling in Fig. 3D seems very diffuse with more in the VZ than SVZ. Lower magnification views should be shown.

A: Well taken. For the TBR2 staining in former Fig. 3D, we used a rather old batch of TBR2 antibody. We have now performed novel staining for TBR2 using a new antibody (and performed blocking with gelatine, as previously reported by Pasca et al. Nature Medicine, 2020), which indeed gave us now a much cleaner and more specific signal. We have now provided a better -and lower magnification- representative example of the TBR2 staining in the ENOs in Figure 3c, which clearly shows the enrichment of TBR2 in the SVZ.

10) Assessing ENOs at later timepoints to more clearly define cortical lamination patterns, the formation of astrocytes, etc. would make the work more useful.

A: We have now included a comparison of the ENOs with COs at day 50 (Figure 3h). These data demonstrate the expression and distribution patterns of SOX5, AUTS2, BRN2, SATB2 and CTIP2 in ENOs, are comparable with those of COs and ENOs, suggesting no difference in neuronal production at later time points between the two protocols. We have now also included analysis at a later timepoint -day 85- to interrogate the formation of astrocytes, which are indeed present in ENOs (and COs) as marked by S100beta, which are intermingled around

CTIP2+ neurons (Figure 3i). Thorough assessment of ENOs at later timepoints is indeed a future focus of our research, while the main focus of the current study was to establish ENOs as a platform to study the early phases of neuroepithelium expansion, reflecting early development.

11) In terms of statistics, some data are obviously not a normal distribution (Fig. 3g, j) but a t-test was used. Although the significance is obvious by eye, please use a nonparametric test for these types of distributions.

A: Thanks for bringing this to our attention. We have now revised our statistics for these quantifications where needed, using a nonparametric Mann-Whitney test.

12) The timelapse cell division data are not convincing with an N of 1 for each group. To truly know what type of cell division is more common, multiple examples or statistical analysis should be performed to provide better evidence. At the very least, multiple examples should be shown (some could go into the supplemental figures).

A: We are sorry for the confusion. The graph provided in former Figure 4k-l (now Figure 5l-m) indeed displayed the average cell division patterns derived from the analysis of the division of multiple cells derived from n=3 different time-lapse imaging experiments. We have now further increased our division analysis and we now also visualized the patterns for each individual cell in the graph in Figure 5l-m. Additionally, we included additional images of cell division examples in Supplementary Figure 12.

Reviewer #2 (Remarks to the Author):

Pagliaro and colleagues develop a method to derive cortical organoids from PSCs with extended neuroepithelium (ENOs) as potential alternative to the currently derived cortical organoids (COs) containing multiple rosettes. The authors propose that such extended neuroepithelial structures mimic more faithfully early human embryonic cortical development, which favors germinal zone expansion, both laterally (length) and vertically (thickness) prior to transition to neurogenic mode.

The authors apply the dual Smad inhibition method as basis for their experiments to dissect factors governing epithelial cell expansion and find that a gradual switch from “full” TGF activation (pluripotent state) to “full” TGF inhibition (neural induction phase) enables lateral progenitor cell expansion on the expense of multiple smaller size rosettes. Authors show decreased neuronal marker expression in ENOs and suggest that by allowing a permissive zone of natural rather than abrupt decrease in TGF signalling (and in the presence of BMP inhibition), extended proliferation phase and delayed neurogenesis occur.

The authors assess cortical identity of ENOs using a number of markers expressed in the cortical epithelium including conventional stem cell markers, positional markers and oRG markers. They show that ENOs are more packed with cells expressing these stem cell and cortical markers when compared to COs – as sign for enhanced cortical specification and delayed neurogenesis.

The authors further show that ENOs exhibit larger ventricular regions delineated by area of luminal perimeter as well as thickness of germinal zone in rosettes / vesicles, and that these measures expand over time, while rosettes in COs decline in size with development. They finally show that apical progenitors bear longer apical-basal processes accompanied by a more densely packed nuclei performing interkinetic nuclear migration in a fashion reminiscent to that occurring in human cortical development in vivo.

This a very interesting study with great potential to the field for a number of reasons. First, a small / simple change made in the protocol makes a big difference in organoid outcome. Second, the change can be readily integrated into other protocols. Third, retaining rosette structure for longer periods is important in attempts to mimic early expansion phases of cortical development. Finally, the notion of a single expanded rosette as the in vitro counterpart of a single cerebral vesicle will allow future arealization studies within organoids.

There are a number of major requests that need to be addressed to strengthen the claims before the work can be considered for publication.

A: We thank the Reviewer for his/her positive evaluation of our manuscript and for highlighting its important aspects. We have found the requests constructive and revised our manuscript accordingly. A detailed response is provided below.

1. Enhanced cortical identity: the authors claim organoids are cortical and show PAX6, EMX2 as cortical and SOX2 as stem cell markers – but they do not show these markers together co-

localized, nor they show at all expression of FOXG1. They should specifically show FOXG1 co-expression with these markers (a. FOXG1/SOX2 for telencephalic stem/progenitors, b. FOXG1/PAX6, FOXG1/EMX2 (or preferably FOXG1/EMX1) for dorsal telencephalic identity. PAX6/EMX1 is also a good combination. These findings should be then compared between COs and ENOs to demonstrate the enhanced cortical identity claimed.

A: We agree with this comment. As a technical drawback, most of the antibodies we used in this study are unfortunately raised in rabbit (including the ones for the novel requested stainings), which makes it difficult to perform co-staining on the same section. These antibodies are in fact the ones validated by their widespread use in multiple previous publications (see e.g. Rosebrock et al. Nature Cell Biology, 2020; Lancaster et al. Nature Biotechnology 2017; Renner et al. EMBO J, 2017; Velasco et al. Nature, 2019).

To satisfactorily address this point, we have taken two different approaches. Firstly, we now used a validated rat SOX2 antibody allowing us to perform co-staining with FOXG1. This indeed confirmed that the identified SOX2+ cells are also largely FOXG1+, confirming the telencephalic stem cell identity of the quantified SOX2+, in both the COs and the ENOs (Figure 4a and supplementary figure 8a). We note that at day 16 the FOXG1 staining is more diffuse, i.e. present both nuclear and to some extent cytoplasmic, whilst at day 24 typical nuclear FOXG1 staining is observed in both ENOs and COs. We have included multiple examples to highlight their telencephalic stem cell identity.

Secondly, to instead confirm the dorsal stem cell identity by FOXG1, EMX1, EMX2, and PAX6 colocalization, we have performed stainings on sequential sections of both ENOs and COs. As now shown in multiple examples in Figure 4j-k and Supplementary Figure 8, this analysis demonstrated that the extended neuroepithelium structures characteristic of the ENOs are indeed robustly and consistently positive for the above mentioned markers, confirming the colocalization and widespread expression of the cortical identity markers. Instead for the COs, the expression of EMX1 and EMX2 was much more varied, presenting with the salt-and-pepper expression pattern as we previously noted for PAX6, and as well expression of these markers tended to be more scattered. Of note, we also performed qRT-PCR for several of these markers comparing the expression in ENOs vs COs, which corroborates our protein staining observations (Figure 4l). Altogether, these novel experiments corroborate the enhanced cortical specification in the ENOs.

2. Increased thickness of germinal zones: similarly, it is important to demonstrate that areas compared between ENOs and COs are composed of cortical stem cells (SOX2) or IP cells (TBR2) based on the above co-expression analyses.

A: The areas compared between the ENOs and the COs are rigorously composed of SOX2 cortical stem cells and/or TBR2+ stem cells. To clarify this aspect, we have added examples of the quantified germinal layers based on SOX2 positivity, until the appearance of TUJ1+ cells, now included as panels in Figure 5e and supplementary figure 11e. Of note, the TBR2 expression pattern is as well highlighted in Figure 3c.

3. The analyses are also required to be demonstrated in more lines (at least one more) - an iPSC line, so that less characterized cells can be tested for robustness, particularly with the dynamic range of endogenous TGF activation level across lines.

A: Well taken. Also in response to a similar comment from Reviewer 1, we now showcase that ENOs can be reliably formed from multiple different lines (H14 and H9 lines, in addition to the already used H1 line), altogether corroborating the robustness of our approach. We have performed rigorous characterization of the ENOs formed from these novel lines, with regard to organoid perimeter, circularity, and area quantifications. We also validated key findings in these lines, such as the enhanced cortical identity as shown by PAX6 staining as well as their inside-out polarity. These extensive novel results are summarized in Figures 1c-h, 2c, and supplementary figures 1i, 3, 4, and 10.

We chose to focus on validating our findings in two additional hESC lines, the H14 and H9 lines in addition to H1, as collectively they represent widely used hESC lines to generate brain organoids (see e.g. Lancaster et al. Nature 2013, Kelava et al. Nature, 2022; Ogawa et al. Cell Rep 2018; Yao et al. Cell Stem Cell 2017; Esk et al. Science, 2020), as well as having these available in-house. Instead, currently, there is no consensus over the use of a single (or a few) iPSC lines across different laboratories to successfully generate brain organoids, which would have required us to perform extensive screening of many iPSC lines, firstly for conventional brain organoid formation, and secondly for ENO formation.

We agree that the dynamic range of the TGF β activation level may vary between lines, as also demonstrated by somewhat varied efficiencies in ENO formation across the H1, H14 and H9 lines (see Figures 1h and 2c, Supplementary Figure 4). In the no SB43 condition, the efficiency of ENO formation is consistently increased. This implies that the ENO protocol can also likely be implemented in less characterized lines, such as iPSC lines, by simply testing both ENO and no SB43 protocols, or slight adjustments of the protocol (i.e. the concentration of SB43, the length and steepness of the gradient).

Finally, to further prove robustness, we considerably increased the number of batches analyzed, the number of organoids analyzed, and performed many more quantifications. Please see also our detailed comment to point 5 of Reviewer 1.

4. The authors should show more complete organoids in their staining (such as Supp Fig. 2) and overall organoid size analysis in ENOs vs COs.

A: Well taken. While generating images of the ENOs poses some technical challenges due to the difficulties to fully preserve the extended and complex neuroepithelial structures during tissue cutting and staining processing, we agree with the importance of showing more complete organoids and have now added multiple examples of whole ENOs and COs for various stainings, including N-cad, ZO-1, PAX6, DCX, TUJ1 etc. (see Figures 1f, 4e, 4i, 5a and Supplementary Figures 3, 5b, 6b, 7a, 7g, 7i, 10, 11a). Finally, we have added movies in which we performed 3D reconstruction imaging of whole ENOs (see Supplementary videos 1-2), where we believe it is easier to visualize the complex 3D structure of the ENOs.

We have now also included extensive measurements of ENOs and COs (across multiple time points and batches) both in terms of organoid perimeter and circularity in supplementary figures 2b-c. Indeed, as we previously showed, organoid perimeters are consistently increased in ENOs relative to COs (explained by its complex structures). We have now quantified the organoid size analysis (organoid area), and we observe a modest -yet significant- increase in organoid area of ENOs vs COs at day 20 and day 25, which may indeed reflect the extended expansion phase (Supplementary Figure 1g). Please note that the perimeter is significantly increased due to the much more complex and convoluted structure of the extended

neuroepithelium in the ENOs, this not necessarily has to be reflected in a proportionate change in organoid area.

While there is no absolute requirement to generate scRNA-Seq datasets for ENOs vs. COs, it is strongly recommended that authors perform such an analysis. This will provide a significant support to the finding that is currently missing due to the use of stainings only in one line only.

A: We agree that it would be valuable to perform scRNA-sequencing of ENOs and COs, which we aim to perform in our future investigations. We believe that, also in consideration of the rather costly nature of scRNA-sequencing, we have provided sufficient support to our findings by corroborating our results now in 3 different hESC lines, by increasing the number of batches and organoids, by adding multiple novel (co-)stainings, confirming the increased cortical identity of ENOs vs COs. Finally, we have now also included a panel of RT-qPCR quantifications of mRNA expression of stem cell/cortical markers comparing ENOs vs COs, further corroborating the enhanced cortical identity observed in ENOs, across 3 different batches with multiple organoids/batches.

Other more minor points:

Fig 1a (scheme) is confusing. Per each treatment box all factors should appear together under one color. Otherwise, it feels that each row is a different treatment (i.e. FGF2 is different treatment than FGF2).

A: Well taken. We now modified the scheme according to this suggestion by showing the different media and the relative components in the same color. We hope that this scheme is now simplified and summarizes more clearly our protocol.

Fig. 3a. Do Day 85 ENOs have any rosettes at all? The image on the right shows only scattered SOX2 cells but no rosettes. That would mean that there is no ventricular lining in late-stage ENOs. Larger area sections of early and late-stage ENOs through time can clarify this.

A: We have now provided better images to clarify this point in figure 3f and supplementary figure 6c-d. We also included polarity images of the organoids shortly after switching to maturation medium, showing the complex polarity rearrangements (supplementary figure 6b). There are small rosettes structures both in the ENOs as well as in the COs at day 85. This is due to the fact that, at this timepoint, most of the neural stem cells differentiate and generate neurons. Please see also our reply to point 4 of Reviewer 1 regarding the polarity of the ENOs.

Fig. 3c. DCX (and SOX2 as well) should be counted within germinal zone only.

A: SOX2 was indeed quantified within the germinal zone. We must admit that we do not fully understand why the DCX should be quantified in the germinal zone only. Newborn neurons are generated in the VZ-like and SVZ-like layers and rapidly migrate on top to generate the neuronal layer. There are indeed only a few DCX positive cells -as expected- in the germinal layers (see and compare figure 3a-b), which we have quantified together with the ones that migrate and locate on top, as this reflects the amount of newly generated neurons

(neurogenesis). For clarity, we have specified in the figure legend that the DCX amount as been quantified in the germinal layers and on top of them.

Fig. 3m: Can authors provide representative staining for COs to assess whether cortical lamination is more efficient in ENOs? If latter is correct then this will support an idea of better cortical diversity in ENOs.

A: We now comparatively assessed ENOs and COs with regard to the panel of neuronal stainings initially performed for ENOs (CTIP2, SATB2, BRN2, AUTS2, SOX5) (now included in figure 3h). This revealed ENOs to be comparable in terms of cellular (neuronal) composition and lamination patterns. This leads us to conclude that generation of ENOs in its current state does not lead to more efficient cortical lamination. We believe that the strength of the protocol lays in the early stages of the organoids, where the neuroepithelium structures are formed as a continuum instead of rosettes. However, we agree that it is of future interest to more thoroughly address lamination patterns with more thorough markers and at even later timepoints.

Fig. 4a bottom left – not clear why ZO-1 faces out. Larger area images should help clarify this.

A: Thanks for the suggestion. We have now indeed included whole-organoid images of ZO-1 staining in ENOs and COs, which now better highlight the difference in polarity between the two protocols (figure 5a, supplementary figure 7a) and we also included supplementary videos 1-2 to better highlight this aspect. In ENOs, the apical side (and apical stem cells) are positioned on the “outside” of the organoid (see also the N-CAD staining in figure 1f, supplementary figures 3, 5b), forming a continued and expanded neuroepithelial structure, whereas COs present with the typical apical lumens throughout the organoid.

Reviewer #3 (Remarks to the Author):

Neural organoids have become a more widely used tool in recent years for modelling human brain development and disease. However, while directed differentiation approaches lead to more homogeneous cortical tissues, tissue architecture is often compromised with somewhat simplistic neural rosettes forming, rather than the elongated, dense telencephalic tissue of the in vivo brain. To overcome this, Pagliaro et al. have devised an interesting and simple method to generate more elongated neuroepithelium, that also reliably forms cortical identity. This is a careful and detailed study demonstrating an important methodological advance that will be vital to the field. I have only a couple comments, as detailed below:

A: We are grateful for the positive and encouraging words of the Reviewer. We have taken in consideration the suggestions of the three Reviewers and carefully revised the paper, and we hope that he/she will appreciate the revised version.

1. The only major comment relates to the polarity of the tissues. Because there is no extracellular matrix provided, such as Matrigel as used in many other brain organoid protocols, the ENOs maintain their apical surface on the external surface of the organoid. This naturally limits the expansion potential upon neurogenesis, as neurons can only go inside. It is also inside-out, compared with the in vivo brain. Finally, it also means the curvature of the neuroepithelium is not as it would be in vivo, since the geometry forces convex regions of curvature, whereas cortical neuroepithelium should have a concave curvature. While parts of the ENOs exhibit this curvature spontaneously (which is very interesting), many regions must instead fold outward. Thus, I wonder if the authors have tried adding Matrigel, either by embedding the ENOs at some stage after initial patterning, or by dissolving Matrigel in the media? This would be a very interesting further direction.

A: The Reviewer is correct regarding the polarity of the tissue; the apical surface is maintained on the external surface of the ENOs until day 25. Until day 25, Matrigel is not added to both the ENO and the "control" standard CO protocol, so we could argue that this is likely due to the emergence of the different tissue architecture linked to the temporal neural induction gradient in the ENOs. We also agree that this constrain limits the capacity of neurogenesis as neurons can only go in the inside, but at the same time this different architecture gives more space to the stem cells.

In a recent preprint from the Knoblich lab testing the effect of Matrigel addition to the formation of brain organoids (Martins-Costa et al., bioRxiv, 2023), the absence of Matrigel determined indeed the formation of organoids with inverted polarity (i.e. with the apical side on the outside of the organoid) and affected the formation of rosettes. Interestingly, however at later time points this does not impair neurogenesis capacity and the overall organoid architecture at later timepoints experiences a spontaneous switch in polarity (i.e. formation of rosettes throughout the organoid). Similarly, since we add Matrigel at day 25 and we switch to maturation medium, the extended neuroepithelium undergoes polarity rearrangement and at later time points ultimately results into internal germinal areas, some of which resembling the rosettes identified in the COs (figure 3a-b, 3f and supplementary figure 6b-d). No changes in neurogenesis were observed between ENOs and COs at day 50 and day 85. We indeed now highlight better this aspect by analyzing the polarity of the ENOs shortly after switching to the

maturation medium (day 35) by ZO-1 staining (supplementary figure 6b) and including lower magnification images of the ENOs at later timepoints (figure 3f, supplementary figure 6c-d).

We do believe that the strength of our methods relies in the generation of an expanded neuroepithelium structure, which constitutes an interesting model to study earlier neurodevelopment with an increased cortical specification.

We agree that it is interesting to test whether this model can be further improved by for instance generating more areas with concave curvature in addition to the ones that are currently spontaneously forming. For this reason, we have taken the Reviewer's suggestion along and tested the effect of Matrigel addition in various concentrations and timing of administration. The results are now presented in supplementary figure 7. If Matrigel is added to the medium prior to neuroepithelium expansion (day 7), ENO structures do not emerge, and organoids form akin to the COs, with the neuroepithelium structures organized as rosettes. We also added Matrigel to the culture medium at a later stage, either when the neuroepithelium structures were at the beginning of their formation (Day 10) or when the structures were already quite expanded (Day 17), and at lower concentrations: either at 0.05% or 0.2%. When Matrigel was added on Day 10, ENOs were not expanding as much as what is usually observed independently of the concentration used. Also here, we reasoned that adding Matrigel at this stage was too early and this would affect the formation of the characteristic expanded structures observed in the ENOs. Collectively, this led us to conclude that the addition of Matrigel at early time points (day 7, 10) is "detrimental" for the formation of the ENOs and does not allow expansion of the neuroepithelium, and this was independent of the concentration used.

When Matrigel was added at later time points (Day 17) and at lower concentration (particularly at 0.05%), this did not interfere with the formation of the expanded neuroepithelium structures. Instead, we noticed an effect on the folding of the neuroepithelium and the generation of multiple areas with concave curvatures. Moreover, the addition of Matrigel in this regiment determined the formation of big, internal lumina in the ENOs. These data therefore provide a proof-of-principle that polarity and architectures of the ENOs can be further improved, and we do believe that indeed this would be an interesting future direction for the advancement ENOs. In fact, we envision testing of different ECM components, with gradient administrations etc., which will be assessed in future studies.

2. It would be important that the authors specify right at the beginning (i.e. in the title and abstract) that the gradient that they are talking about is temporal. As it stands, the wording could lead to initial confusion on the part of the reader, since in development gradients are usually spatial or spatio-temporal.

A: We appreciate the suggestion, and we agree that the title was confusing regarding the type of gradient used in the presented protocol. We now changed the wording in the title and in the abstract to make it clearer to the reader.

REVIEWERS' COMMENTS

Reviewer #1 (Remarks to the Author):

In this revised manuscript from Pagliaro et al, the authors assess the effects of using a stepwise transition of morphogens and small molecule inhibitors during neural patterning in embryoid bodies to generate brain organoids. The authors find that the organoids contain an “inside-out” continuous neuroepithelium with dramatically different shape and organizational properties than traditional cortical organoids. The authors have addressed many of the original reviewer concerns in this revision, including adding additional human pluripotent stem cell lines, quantification, example images, and others. However, a few minor concerns remain that should be addressed:

1) The apical-out nature of the organoids is now shown by images and mentioned in the text; however, this is a crucial point about ENOs that needs to be highlighted earlier in the results and perhaps in the abstract. Please also label the “outside” of the ENO’s Figures 2f, 3b,c, and 4a,f,j.

2) To this same point, it is intriguing that the ENOs remain apical-out presumably until the introduction of ECM (Matrigel). The Sasai method (SFEBq) method generates apical out brain organoids that then become apical-in after removal of KOSR containing media (Kadoshima et al, 2013 Figure 2A, Watanabe et al, 2017 Figure 1A). Your results suggests that your ENO cells are not undergoing apical constriction until ECM administration.

Interestingly, apical constriction seems dependent on the level of BMP:

<https://doi.org/10.1002/bdra.23052>. The authors may want to mention this.

3) What was the criteria for ENO efficiency (Figure 1h, Figure 2c, etc)? Was it based on immunostaining, elongated morphology, apical-out polarity, or some other criterion? This would be helpful to put in the methods section.

4) The fact that complete loss of SB43 leads to the same phenotype as graded TGF-beta/SB43 transition and ENO efficiency is higher with reduced SB43 is concerning. If true, is SB431542 superfluous? Do the SB43 organoids have the same cell type identity as the

ENOs? Were these immunostained? This finding would be unusual as SB43 has been shown to be indispensable for previous dual-SMAD differentiation protocols. This point merits more discussion.

Reviewer #2 (Remarks to the Author):

The revised study by Pagliaro et al is now a much thorough study and has rigorously addressed the major points raised.

The authors must dedicate a section (Results but mainly Discussion) to rigorously discuss the issue of organoid polarity. Since developmental cell composition of COs and ENOs seems similar, they should now provide their detailed view (as much as space allows) on the differences, potential advantages and limitations of ENOs vs COs in light of their protocol and in context with the expectation of the reader to be able to readily realize inside out polarity structure and anatomical region / cell type composition. They should include an illustration with VZ, SVZ/oSVZ, CP areas demarcated. Also, adding a deviation route from ENOs to COs depending on Matrigel should be added.

Reviewer #3 (Remarks to the Author):

The revised manuscript is greatly improved and has addressed the previous concerns satisfactorily. I only have a couple very minor comments:

Line 113, 127, 951

Analogue should be replaced with analogous

Line 301

“Luminal perimeter” should be changed to “apical perimeter” for consistency and for the sake of precision I would prefer a term such as apical length since in the ENOs there might be one or several linear neuroepithelia to which the term perimeter does not apply

REVIEWERS' COMMENTS

Reviewer #1 (Remarks to the Author):

In this revised manuscript from Pagliaro et al, the authors assess the effects of using a stepwise transition of morphogens and small molecule inhibitors during neural patterning in embryoid bodies to generate brain organoids. The authors find that the organoids contain an “inside-out” continuous neuroepithelium with dramatically different shape and organizational properties than traditional cortical organoids. The authors have addressed many of the original reviewer concerns in this revision, including adding additional human pluripotent stem cell lines, quantification, example images, and others. However, a few minor concerns remain that should be addressed:

A: We thank the Reviewer for the positive evaluation of our manuscript.

1) The apical-out nature of the organoids is now shown by images and mentioned in the text; however, this is a crucial point about ENOs that needs to be highlighted earlier in the results and perhaps in the abstract. Please also label the “outside” of the ENO’s Figures 2f, 3b,c, and 4a,f,j.

A: Well taken. We now mention the apical-out nature of the ENOs already in the abstract and earlier in the text (line 25, lines 124-126). We furthermore labeled the outside of the ENOs in the mentioned Figure panels as requested.

2) To this same point, it is intriguing that the ENOs remain apical-out presumably until the introduction of ECM (Matrigel). The Sasai method (SFEBq) method generates apical out brain organoids that then become apical-in after removal of KOSR containing media (Kadoshima et al, 2013 Figure 2A, Watanabe et al, 2017 Figure 1A). Your results suggests that your ENO cells are not undergoing apical constriction until ECM administration. Interestingly, apical constriction seems dependent on the level of BMP: <https://doi.org/10.1002/bdra.23052>. The authors may want to mention this.

A: Agreed. ENOs cells seem not to undergo apical constriction until the addition of Matrigel. Presumably apical constriction is regulated by multiple signaling pathways and in different brain organoid protocols it is induced by different means. We now included in the discussion a section in which we discuss the possible correlation between TGF- β , BMP and ECM signaling with ENOs outside-in morphology.

3) What was the criteria for ENO efficiency (Figure 1h, Figure 2c, etc)? Was it based on immunostaining, elongated morphology, apical-out polarity, or some other criterion? This would be helpful to put in the methods section.

A: Thanks for the suggestion. We now included a description on the criteria for ENO formation efficiency in the methods section. Briefly, we defined ENOs as successfully formed based on brightfield images, by analysing them from day 14 onwards. At this stage the difference in morphology between COs and ENOs becomes apparent (Figures 1b, 2b and Supplementary Figures 1c, 2a, 4a-b). We judge ENOs formation by the presence of convoluted and elongated

structures, as opposed to COs, instead displaying the typical spherical organoid shape. Moreover, ENOs show to be more translucent compared to COs. This assessment is corroborated by immunostainings of the assigned ENOs, demonstrating the presence of expanded neuroepithelium structures, displaying all the features typical of ENOs.

4) The fact that complete loss of SB43 leads to the same phenotype as graded TGF-beta/SB43 transition and ENO efficiency is higher with reduced SB43 is concerning. If true, is SB431542 superfluous? Do the SB43 organoids have the same cell type identity as the ENOs? Were these immunostained? This finding would be unusual as SB43 has been shown to be indispensable for previous dual-SMAD differentiation protocols. This point merits more discussion.

A: Indeed, the complete loss of SB43 does lead to a similar phenotype as with the graded TGF- β /SB43 transition. These organoids are morphologically similar and moreover show correlation also in the mRNA expression of specific markers as shown in Figure 4I. Solely in the context of ENOs formation, SB43 seems to be superfluous; however, we did not investigate this in the context of the COs protocol. Published protocols to generate cortical organoids mostly rely on dual-SMAD inhibition; however, the concentration of these inhibitors varies substantially across protocols (for example for SB43 from 1 (PMID: 32142682) to 10 μ M (PMID: 28445465). Additionally, protocols employing dual SMAD inhibition rely on findings from cortical specification in 2D cultures (PMID: 19252484), and this has been adopted to 3D culture. We are not aware of studies directly comparing the effect of a single vs dual SMAD inhibition on the specification of 3D brain organoids. It may very well be that the signaling pathway activities in 3D differ from those observed in 2D culture, which is a well-known phenomenon in 2D-3D in vitro culture. Furthermore, it is evident that proper 3D architecture is very important for cellular identity, as also recently shown (PMID: 37802039), and this may overrule the importance of two SMAD inhibitors, especially in the ENOs. This is definitely an interesting point of discussion, and as suggested, we added a paragraph about this in the Discussion.

Reviewer #2 (Remarks to the Author):

The revised study by Pagliaro et al is now a much thorough study and has rigorously addressed the major points raised.

A: Thanks!

The authors must dedicate a section (Results but mainly Discussion) to rigorously discuss the issue of organoid polarity. Since developmental cell composition of COs and ENOs seems similar, they should now provide their detailed view (as much as space allows) on the differences, potential advantages and limitations of ENOs vs COs in light of their protocol and in context with the expectation of the reader to be able to readily realize inside out polarity structure and anatomical region / cell type composition. They should include an illustration with VZ, SVZ/oSVZ, CP areas demarcated. Also, adding a deviation route from ENOs to COs depending on Matrigel should be added.

A: We now added a paragraph in the Discussion concerning the organoid polarity and the

*advantages and limitations of ENOs vs COs. As suggested, we also now included an illustration of the effects of Matrigel on ENOs to COs which includes the demarcation of difference zones/cell types in these organoids (new **Supplementary Figure 8**).*

Reviewer #3 (Remarks to the Author):

The revised manuscript is greatly improved and has addressed the previous concerns satisfactorily. I only have a couple very minor comments:

Line 113, 127, 951

Analogue should be replaced with analogous

A: Thanks. We changed this.

Line 301

“Luminal perimeter” should be changed to “apical perimeter” for consistency and for the sake of precision I would prefer a term such as apical length since in the ENOs there might be one or several linear neuroepithelia to which the term perimeter does not apply

A: Thanks. We now modified the text accordingly also to maintain consistency throughout the manuscript.